# AN OVERLOOKED INGREDIENT FOR RECTIFIED FLOW: REAL SAMPLES

## ABSTRACT

Rectified flow is a generative model that learns smooth transport mappings between two distributions through an ordinary differential equation (ODE). Unlike diffusion-based generative models, which require costly numerical integration of a generative ODE to sample images with state-of-the-art quality, rectified flow uses an iterative process called reflow to learn smooth and straight ODE paths. This allows for relatively simple and efficient generation of high-quality images. However, rectified flow still faces several challenges. 1) The reflow process requires a large number of generated pairs to preserve the target distribution, leading to significant computational costs. 2) Since the model is typically trained using only generated image pairs, its performance heavily depends on the 1-rectified flow model, causing it to drift away from the real data.

In this work, we expose the limitations of the original rectified flow and propose a novel approach that incorporates real images into the reflow procedure. By preserving the ODE paths toward real images, our method effectively reduces reliance on large amounts of generated data. Instead, we demonstrate that the reflow process can be conducted efficiently using a much smaller set of generated and real images. In CIFAR-10, we achieved significantly better FID scores, not only in one-step generation but also in full-step simulations, while using only 7.2% of the generative pairs compared to the original method. Furthermore, our approach induces straighter paths and avoids saturation on generated images during reflow, leading to more robust ODE learning while preserving the distribution of real images.

## 1 INTRODUCTION

Rectified flow (Esser et al., 2024; Liu et al., 2023; Lee et al., 2024; Li et al., 2024; Lee et al., 2023) demonstrates state-of-the-art image generation with fewer sampling steps than diffusion models (De Bortoli et al., 2021; Vargas et al., 2021; Song et al., 2020b; Ho et al., 2020; Tzen & Raginsky, 2019). K-rectified flow involves $k$ reflow steps that make ODE paths smooth and straight (Liu et al., 2022). This allows the model to generate high-quality images simply and efficiently in just one or a few steps. Intriguingly, *all rectified flow models* (Flux, SD3, and AuraFlow) achieve state-of-the-art quality with *1-rectified flow* and require about 30 NFEs (number of function evaluations) (Esser et al., 2024).

In this paper, we find the pitfalls of k-rectified flow because of its reflow procedure and propose to use real images and their inversions as solutions. 1) The flow drifts away from the real distribution. For example, to train a 2-rectified flow, random starting noises and their corresponding generated images through the 1-rectified flow supervises the ODE paths to be straight, i.e., the generated images themselves are reused as training data. Using the real samples keeps the supervision toward the real distribution. 2) The number of required fake samples is very large. Using the real samples and their conic neighbor outperforms with much fewer samples. 3) Reflow procedure lowers image quality of full-step generation. Our method improves image quality in all 1-step, few-step, and full-step generation.

As a result, we successfully demonstrate better performance than existing k-rectified flow models. On CIFAR-10, we reduce the FID (Fréchet Inception Distance, (Heusel et al., 2017)) of the existing 2-rectified flow from 12.21 to 5.98 while using only 7.2% of the generative pair, and show that

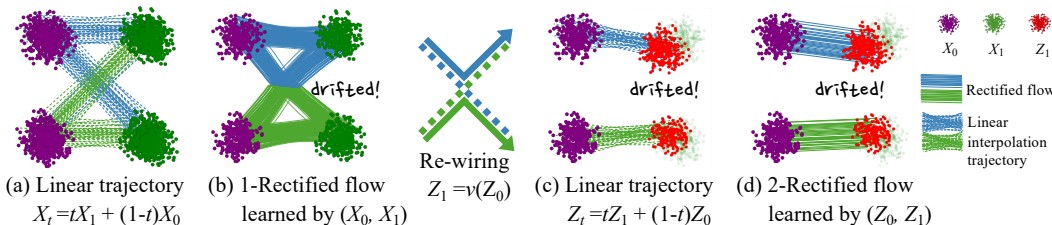

(a) Linear trajectory
$X_t = tX_1 + (1\text{-}t)X_0$

(b) 1-Rectified flow
learned by $(X_0, X_1)$

Re-wiring
$Z_1 = v(Z_0)$

(c) Linear trajectory
$Z_t = tZ_1 + (1\text{-}t)Z_0$

(d) 2-Rectified flow
learned by $(Z_0, Z_1)$

Figure 1: **Problem of rectified flow.** (a) By randomly pairing data $X_0 \sim \pi_0$ and $X_1 \sim \pi_1$, linear interpolation trajectories are defined. (b) The 1-rectified flow connects $X_0$ and $X_1$ with a learned velocity field which is potentially inaccurate. After the 1-rectified flow, the trajectories are rewired to avoid crossing. (c) The trajectories for reflow are defined as linear interpolation trajectories between $Z_0$ and the generated $Z_1 = v(Z_0)$. Note that $Z_1$ is drifted away from $\pi_1$. (d) Consequently, the 2-rectified flow has velocity field drifted away from $X_1$.

the curvature of the ODE paths became straighter. We experimentally and numerically illustrate the issues of using generated pairs from the original k-rectified flow as training data, and propose conic-reflow to address these problems. We also introduce a new method for calculating curvature, which explains the time distribution sampling method. We provide various ablation studies and show that even with simple fine-tuning, the performance of existing k-rectified flow can be significantly improved.

To the best of our knowledge, this is the first paper to advance k-rectified flow, and we believe that this paper will flourish rectified flow literature.

## 2 RECTIFIED FLOW

Rectified flow (Liu et al., 2022) is a generative model that solves an ordinary differential equation (ODE) to induce a transition trajectory between two given data distributions $\pi_0$ and $\pi_1$. Data $X_0 \sim \pi_0$ and $X_1 \sim \pi_1$ define linear trajectories $X_t = (1 - t)X_0 + tX_1$ for $t \in [0, 1]$ as illustrated in Figure 1a. Then, a rectified flow $v$ is an ODE on time $t$ parameterized by $\theta$:

$$\frac{dZ_t}{dt} = v_\theta(Z_t, t) := \frac{1}{t}(Z_t - \mathbb{E}[(X_1 - X_0)|X_t = Z_t]). \tag{1}$$

We omit $\theta$ for brevity. Liu et al. (2022) propose a simplified mean squared error (MSE) loss for an ODE neural network to train velocity field $v : \mathbb{R}^n \to \mathbb{R}^n$ as follows:

$$\hat{\theta} = \arg\min_\theta \mathbb{E}\left[\|X_1 - X_0 - v(tX_1 + (1 - t)X_0, t)\|^2\right], \quad \text{with } t \sim \text{Uniform}([0, 1]). \tag{2}$$

In image generation tasks, $X_0 \sim \pi_0$ and $X_1 \sim \pi_1$ are latent noises and real images, sampled from Gaussian distribution and data distribution, respectively. Once the model has learned the velocity field, the Rectified Flow rewires the trajectories in a non-crossing manner, as depicted in Figure 1b. It constitutes a 1-rectified flow model denoted by $\mathbf{Z} = \texttt{RectFlow}((X_0, X_1))$.

The k-rectified flow model learns a straighter sampling trajectory by repeating *reflow procedure* k times as follows. Following $\mathbf{Z}^k$ from $Z_0^k$ induces a generated pair $(Z_0^k, Z_1^k)$ where $(Z_0^0, Z_1^0) = (X_0, X_1)$. It redefines the linear interpolation trajectory $\mathbf{Z}_t^{k+1} = (1 - t)Z_0^k + tZ_1^k$ for $t \in [0, 1]$ as shown in Figure 1c. Then, fine-tuning a velocity field $v$ using equation 2 with $(Z_0^k, Z_1^k)$ instead of $(X_0, X_1)$ constitutes $\mathbf{Z}^{k+1} = \texttt{RectFlow}((Z_0^k, Z_1^k))$

According to optimal transport theory, (Villani, 2021; 2009; Figalli & Glaudo, 2021; Flamary et al., 2016) coupling the generated pairs $(Z_0, v(Z_0))$ ensures that the interpolation trajectory preserves the marginal distributions of the original and target domains, as well as the linear interpolation trajectory between them (Kurtz, 2011; Ambrosio et al., 2008). $k$-rectified flow has superior quality of few-step sampling by straighter sampling trajectory as shown in Figure 1d.

## 3 IMPROVED TECHNIQUES FOR RE-FLOW STEP

In this section, we discuss the fake pairs used in the original rectified flow and their problems. Then, we introduce real pairs and balanced-conic reflow which directly supervise the flow to reach the real data distribution. Finally, we provide detailed training configurations.

### 3.1 REFLOW STEPS DRIFT THE FLOW AWAY FROM THE REAL DISTRIBUTION.

As shown by Liu et al. (2022), the trajectory $Z_t^k$ between the generated pairs in the reflow process becomes smoother and straighter with each iteration of reflow, due to the fact that the ODE induce a deterministic smooth solution while preserving the same marginal distribution as the original trajectory. This straightened path is essential for generating high-quality images with small number of sampling steps rather than SDE based generative models. (Ho et al., 2020; Song et al., 2020b;a)

We use subscript $F$ to denote the **fake pairs** from the original rectified flow as follows:

$$(Z_{0,F}^k, Z_{1,F}^k) := (Z_0^k, Z_1^k), \quad \text{where } Z_0^k \sim \pi_0 \text{ and } Z_1^k = v(Z_0^k).^1 \tag{3}$$

To simplify notation and avoid confusion, we will denote the fake pair as $(Z_{0,F}, Z_{1,F})$ when we do not need to consider the reflow step and, denote the $k$-th order of the rectified flow as $(Z_{0,F}^k, Z_{1,F}^k)$.

Interestingly, the $k$-rectified flow underperforms the $(k-1)$-rectified flow in terms of image quality. This is obvious because the fake samples have lower quality than the real samples, as we do not have the optimal generative model (Figure 3a). Figure 2 illustrates a toy example of the distribution shift due to the difference between real and fake samples. The drift accumulates over the subsequent reflow steps and harms image quality of higher-order rectified flows.

Figure 2 illustrates faithfulness and continuity[2] of an 2-rectified flow on both fake samples and real samples. As expected, fake images are roughly reconstructed by inversion and generation following the 2-rectified flow, i.e., $Z_1 \simeq v(v^{-1}(Z_1))$. Also, an inversion of a fake sample and its perturbation land at similar images, i.e., $v(v^{-1}(Z_1)) \simeq v(v^{-1}(Z_1) + \varepsilon z)$ we use $\varepsilon = 0.05$. In contrast, real images lose the main object by inversion and generation following the 2-rectified flow, i.e., $X_1 \neq v(v^{-1}(X_1))$. Furthermore, real images are vulnerable to perturbation on their inversion, i.e., $v(v^{-1}(X_1)) \neq v(v^{-1}(X_1) + \varepsilon z)$.

To evaluate the faithfulness of a rectified flow to a dataset $X$, we measure the error between the sample and its reconstruction (Candes et al., 2006; Ravishankar et al., 2019) via the flow:

$$L_2^{\text{recon}}(X) = \mathbb{E}_{x \sim X} \left[ \|x - v(v^{-1}(x))\|_2 \right]. \tag{4}$$

Instead of Lipschitz continuity, we practically evaluate the continuity of a rectified flow near samples from a dataset $X$ by measuring a perturbed reconstruction error:

$$L_2^{\text{p-recon}}(X, \varepsilon) = \mathbb{E}_{x \sim X, z \sim \pi_0} \|x - v(v^{-1}(x) + \varepsilon z)\|_2, \tag{5}$$

where $\varepsilon$ is the strength of perturbation. The lower perturbed reconstruction error near the real samples $L_2^{\text{p-recon}}(X_1)$ indicates the more continuous generative model near the real samples.[3]

Figure 3b compares $L_2^{\text{recon}}$ of real and fake samples, and $L_2^{\text{p-recon}}$ near real and fake samples. $L_2^{\text{recon}}$ is higher at the real samples than the fake samples. It indicates that the 2-rectified flow drifts away from the real samples. Furthermore, $L_2^{\text{p-recon}}$ is lower near the fake samples than the real samples. It indicates that the 2-rectified flow suffer from crossing between real samples.

Critically, the undesirable drift in real images continues to have an impact as more reflow steps are applied. It is an innate phenomenon because the supervision from a shifted distribution does not steer the flow toward the real distribution. We provide a solution in the next subsections.

### 3.2 REAL PAIR

The previous subsection has unveiled the pitfall of supervision using fake pairs: samples from the domain distribution, e.g., Gaussian, and their codomain following a rectified flow. Instead of the fake pairs, we propose to use the real samples and their inverse following a reverse rectified flow, defined by

$$\text{Real pair } (Z_{0,R}, X_1) := (v^{-1}(X_1), X_1), \quad \text{with } X_1 \sim \pi_1, \tag{6}$$

---

[1]For the sake of notational simplicity, we denote the forward generation process at the t-sampling step, $X_0 + v_t(X_t, \cdot) \, dt := v(X_0)$ and backward process $X_1 + v_t^{-1}(X_t, \cdot) \, dt := v^{-1}(X_1)$, where $v^{-1} = -v$.

[2]We use the notion of continuity as in Lipschitz continuity: the generated images should be similar with the similar latents.

[3]We measure the reconstruction and perturbed reconstruction error with 1-step Euler sampling.

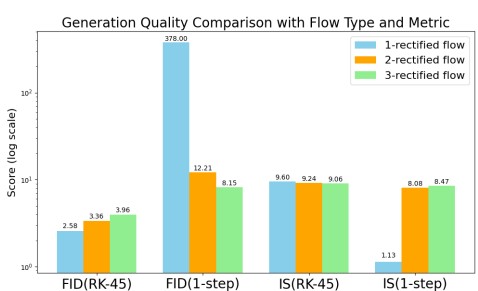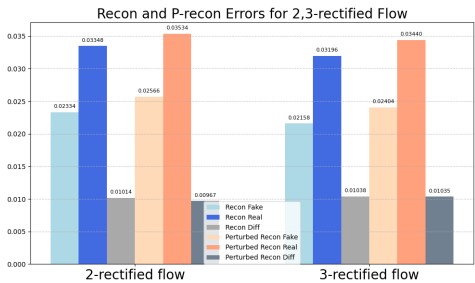

Figure 2: **2-rectified flow overfits fake samples.** Following the reverse and forward 2-rectified flow, fake images inherently return at similar images with or without perturbation at $\pi_0$. In contrast, real images return at different images and it is worse with perturbation, implying the overfitting.

(a) Higher-order rectified flow degrades multi-step quality.

(b) Since the reflow process uses only fake pairs, discrepancies occur between the reconstruction errors of real and fake images.

Figure 3: **Rectified flow suffer from drifts.** (a) The reflow process improves 1 step sampling quality but degrades multi-step sampling quality. (b) discrepancies occur between the reconstruction errors of real and fake images.

where $v$ is the 1-rectified flow and we abuse the term *real pair* although the $Z_{0,R}$ is not real. As in the original rectified flow, where it was optionally provided, it is safe and easy to use reverse rectified flow without stochasticity because it inherently produces a deterministic solution, and using real images does not contradict the original purpose because the noise $v^{-1}(X_1)$ is generated using $v^{-1}$.

Please refer to Figure 4 to visually understand the definition of real pairs compared to fake pairs. To avoid confusion, from now on, we will refer to $(X_0, v(X_0))$ as a *fake pair* (generated pair), where $v(X_0)$ is a fake (generated) image, and $(v^{-1}(X_1), X_1)$ as a *real pair*, where $X_1$ is a real image.

### 3.3 CONIC REFLOW

Building upon the basic pairing of real samples with their reverse noises, we introduce conic reflow, which expands their influence on the domain distribution to their neighboring areas, as shown in Figure 4b. When we train the model, we use spherical linear interpolation (Slerp) between the reverse noise $Z_{0,R}$ and a randomly sampled noise $\epsilon \sim \mathcal{N}(0, I)$ with the interpolation ratio $\zeta$:

$$\text{Slerp}(Z_{0,R}, \epsilon, \zeta) = \frac{\sin((1-\zeta)\phi)}{\sin(\phi)} Z_{0,R} + \frac{\sin(\zeta\phi)}{\sin(\phi)} \epsilon, \tag{7}$$

where $\phi = \arccos(Z_{0,R} \cdot \epsilon)$ denotes the angle between $Z_{0,R}$ and $\epsilon$. Then we define a conic inverse from a real sample $X_1$:

$$\text{Conic}(X_1, \epsilon, \zeta, t) = tX_1 - (1-t)\text{slerp}(Z_{0,R}, \epsilon, \zeta), \tag{8}$$

where $Z_{0,R} = v_\theta^{-1}(X_1)$, and $t \in [0, 1]$. During training, we sample $\epsilon$ and $\zeta$ multiple times to let the target flow stochastically cover the nearby domain. As the collection of the paths over multiple iterations looks like a cone, we name our method as *conic reflow*. The schedule of interpolation weight $\zeta$ is deferred to § 3.5. Our training objective with conic reflow is:

$$\hat{\theta} = \arg\min_\theta \int_0^1 \mathbb{E}\left[ w_t \|X_1 - \text{slerp}(Z_{0,R}, \epsilon, \zeta) - v_\theta(\text{Conic}(X_1, \epsilon, \zeta, t))\|^2 \right] dt \tag{9}$$

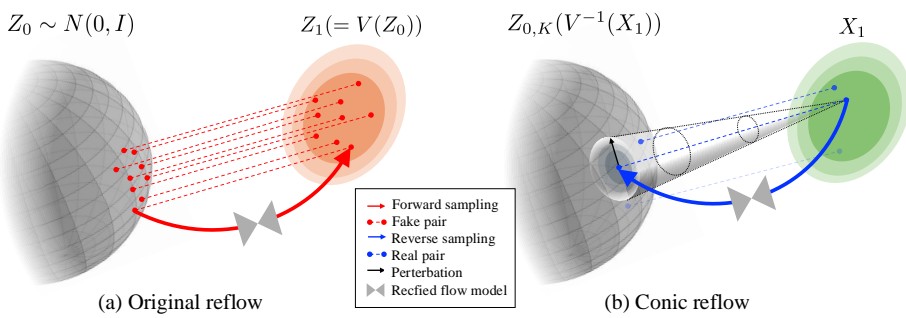

Figure 4: **Illustration of original fake pairs and our real pairs.** (a) The original rectified flow supervises 2-rectified flow with fake pairs $(Z_0, v_\theta(Z_0))$. (b) Our conic reflow supervises 2-rectified flow with real pairs $(v_\theta^{-1}(X_1), X_1)$ and their conic neighbors.

$$\text{where} \quad t \sim \exp([0,1]), \quad \epsilon \sim \mathcal{N}(0, I), \quad \zeta \sim \text{slerp schedule}([0,1]). \tag{10}$$

$$w_t \text{ is weighting function (default=1).} \tag{11}$$

Slerp is preferred for noise space interpolation in generative models, as it preserves results on the Gaussian hypersphere, ensuring consistent magnitude, and smooth transitions (Wang & Golland, 2023; Jang et al., 2024). Inherently, the rectified flow learned with our method still avoids crossing in single cone. Furthermore, slerp with Gaussian noise has the meaning of resolving the discrepancy between the inversion of real samples and domain distribution due to numerical error and suboptimal learned flow by pushing the inversion toward the domain distribution.

## 3.4 BALANCED CONIC RECTIFIED FLOW

We design a new reflow procedure which consists of our conic reflow and the original reflow. For each training iteration, we design different training schemes for real pairs and fake pairs. We alternate between conic reflow steps with real pairs $(Z_{0,R}, X_1)$, and original reflow steps with fake pairs. It encourages the trajectories to head toward the exact real distribution while fake pairs ensures the entire domain distribution to receive supervision.

---

**Algorithm 1:** Rectified flow with balanced conic reflow: Full Algorithm

---

**Procedure: Z** $=$ RectFlow$(X_0, X_1)$
**Inputs:** Draws from a coupling $(X_0, X_1)$ of $\pi_0$ and $\pi_1$; velocity model $v_\theta : \mathbb{R}^d \to \mathbb{R}^d$ with parameter $\theta$.
**Training:** $\hat{\theta} = \arg\min_\theta \mathbb{E}\left[\|X_1 - X_0 - v_\theta(tX_1 + (1-t)X_0, t)\|^2\right]$, with $t \sim \text{Uniform}([0,1])$.
**Sampling:** Draw $(Z_{0,F}Z_{1,F})$ following $dZ_{t,F} = v_{\hat{\theta}}(Z_{t,F}, t)dt$ starting from $Z_{0,F} \sim \pi_0$ and $(Z_{0,R}, X_1)$
   following $dZ_{t,R} = v_{\hat{\theta}}^{-1}(Z_{t,R}, t)dt, \quad X_1 \sim \pi_1$.

**Balanced conic reflow:** Define Reparing step $= \mathcal{T}$, $\chi_{\text{fake}}$, $\chi_{\text{kernel}}$ $cnt = 0$
For total training step $\mathbb{N}$:
—If cnt $== \mathcal{T}$:
——Repairing New Real pair $(Z_{0,R}, X_1) \leftarrow (v_{\theta_{conic}}^{-1}(X_1), X_1)$
——cnt $= 0$
—else:
——**Training:**$\hat{\theta}_{conic}$

$$\arg\min_\theta \mathbb{E}\left[\left\|\chi_{\text{fake}} \cdot \left(\dot{Z}_{t,F} - v_\theta(Z_{t,F})\right) + \chi_{\text{real}} \cdot \left(X_1 - \text{slerp}(Z_{0,R}, \epsilon, \zeta) - v_\theta\left(\text{Conic}(X_1, \epsilon, \zeta, t)\right)\right)\right\|^2\right]$$

**Distill (optional):** Learn a neural network $\hat{T}$ to distill the $k$-rectified flow, such that $Z_{1,F}^k \approx \hat{T}(Z_{0,F}^k)$.

---

Fake Pairs: For fake pairs, we proceed with the reflow process exactly as it was done in the original rectified flow model. Its training objective is as follows:

$$\hat{\theta} = \arg\min_\theta \mathbb{E}\left[\|Z_{1,F} - Z_{0,F} - v_\theta(tZ_{1,F} + (1-t)Z_{0,F})\|^2\right], \quad \text{with } t \sim \exp([0,1]). \tag{12}$$

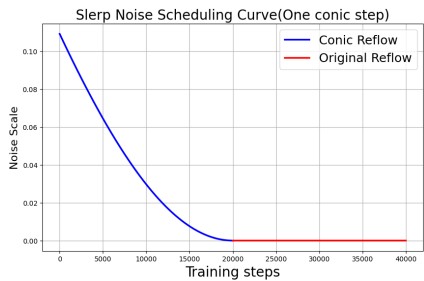
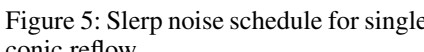

Figure 5: Slerp noise schedule for single conic reflow.

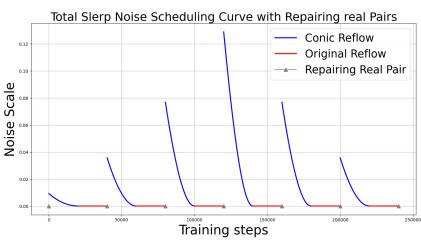

Figure 6: Example of total balanced conic flow training schedule (up to 240K steps).

The entire training objective of our method is as follows for given fake pair$(Z_{0,F}, Z_{1,F})$and real pair$(Z_{0,R}, X_1)$:

$$\min_v \int_0^1 \left[ \left\| \chi_{\text{fake}} \cdot \left( \dot{Z}_{t,F} - v_\theta(Z_{t,F}) \right) + \chi_{\text{real}} \cdot \left( X_1 - \text{slerp}(Z_{0,R}, \epsilon, \zeta) - v_\theta\big(\text{Conic}(X_1, \epsilon, \zeta, t)\big) \right) \right\|^2 \right] dt \tag{13}$$

where $\chi_{\text{fake}}$ and $\chi_{\text{real}}$ are indicator functions for given index subsets $U_{\text{real}}$ and $U_{\text{fake}}$ such that $U_{\text{real}} \cup U_{\text{fake}} = \mathbb{N}$. Then, for $i \in \mathbb{N}$:

$$\chi_{\text{fake}} = \begin{cases} 1 & \text{if } i \in U_{\text{fake}} \\ 0 & \text{else} \end{cases}, \qquad \chi_{\text{real}} = \begin{cases} 1 & \text{if } i \in U_{\text{real}} \\ 0 & \text{else.} \end{cases} \tag{14}$$

For the entire training step $\mathbb{N}$, Conic, $\zeta$, $\epsilon$, and $w_t$ follow equation 10 and equation 11.

### 3.5 DETAILED TRAINING SCHEMES

In this section, we provide a more detailed explanation of our proposed training schemes, including visualizations of the slerp scheduling.

**Slerp noise schedule** Figure 5 shows a single conic reflow noise schedule in training steps where the maximum noise is 0.13. Each conic is trained to progressively reduce the noise scale over time. Specifically, the Slerp noise schedule $\zeta$ is defined as proportional to slerp schedule$(\zeta) \propto 1 - \frac{\zeta^2}{1+\zeta^2}$, $\zeta \in [0,1]$. This design ensures that the noise starts at a large value and gradually decreases to zero as training progresses. This choice aligns with the intuition from traditional diffusion models(Ho et al., 2020), where noise is progressively reduced to refine the generated sample towards realism.

Figure 6 illustrates an example of single conic reflow adjustment during training. For visualization convenience, the figure assumes 240K total steps and visualizes the Slerp noise. In actual training, the noise increases linearly from 0.006 to 0.13, then decreases symmetrically back to 0.006 over 600K steps.

**Exponential time distribution** Additionally, we employ an exponential based time distribution[4](Lee et al., 2024) for linear trajectory $\mathbf{Z}_t$. For more detailed information, see Appendix A.

## 4 EXPERIMENTS

We conducted experiments to evaluate the effectiveness of our Balanced Conic Reflow method. Our findings demonstrate that: Superiority over original reflow in terms of (1) Quality of the results, (2) Straightness of the flow, and (3) Mitigation of Distribution Shift , as well as (4) Ablation study, (5) Improvements through fine-tuning a pre-trained vanilla rectified flow model and lastly, (6) Generalization to Other Datasets.

**Experimental setup** Most of our experiments are conducted on CIFAR-10 (Krizhevsky et al., 2009). The IVD, curvature, reconstruction, and perturbed($0.05\varepsilon$, 1-step) reconstruction error values reported were computed using 10,000 random samples, with the expectation taken over these samples. We employ Scipy's RK45(Virtanen et al., 2020), a 5(4) Runge-Kutta method with adaptive step

---

[4]More explicitly, we use $p_t(u) \propto \exp(au) + \exp(-au)$ on $u \in [0,1]$ with $a = 3$

| Method | NFE (↓) | IS (↑) | FID (↓) |
|---|---|---|---|
| **One-Step Generation (Euler solver, N=1)** | | | |
| 1-Rectified Flow | 1 | 1.13 (9.08) | 378 (6.18) |
| *2-Rectified Flow* | | | |
| Original (+*Distill*) | 1 | 8.08 (9.01) | 12.21 (4.85) |
| **Ours** (+*Distill*) | 1 | **8.79 (9.11)** | **5.98 (4.16)** |
| *3-Rectified Flow* | | | |
| Original (+*Distill*) | 1 | 8.47 (8.79) | 8.15 (5.21) |
| **Ours** (+*Distill*) | 1 | **8.84 (8.96)** | **5.48 (4.68)** |
| **Full Simulation (Runge–Kutta (RK45), Adaptive N)** | | | |
| 1-Rectified Flow | 127 | **9.60** | **2.58** |
| *2-Rectified Flow* | | | |
| Original | 110 | 9.24 | 3.36 |
| **Ours** | 104 | **9.30** | **3.24** |
| *3-Rectified Flow* | | | |
| Original | 104 | 9.01 | 3.96 |
| **Ours** | 98 | **9.14** | **3.70** |

Table 1: Comparison of 2- and 3-Rectified Flows for Original vs Ours using one-step generation and full simulation. Comparison with diffusion-based models is provided in Appendix J.

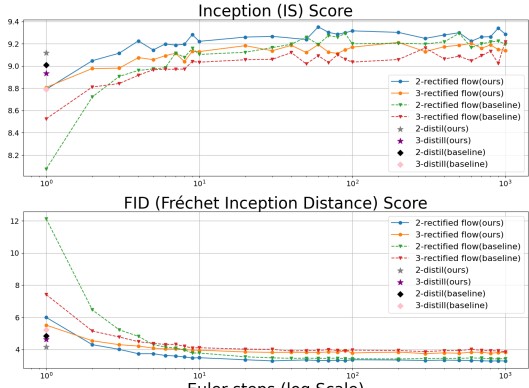

Figure 7: Generation quality for different Euler steps.

size and step count determined by specified tolerances, following the same parameters (Song et al., 2020b). Further details on the training configurations are provided in Appendix I.

## 4.1 IMAGE QUALITY

Our method achieves better FID and IS scores across all sampling steps, i.e., 1-step, few-step, and full-step simulations, as shown in Table 1 and Figure 7. Notably, we use only 300K fake pairs compared to the 4M fake pairs used in the original rectified flow, highly highlighting the efficiency of our approach. Furthermore, our method achieves better generation quality in RK sampling, despite having lower NFE than the baseline. This indicates reduced drift away from real images in the reflow process. This suggests the importance of real pairs, even though we use a relatively small number (60K) of them.

Additionally, even when applying the same distillation methods used in the original rectified flow (Liu et al., 2022), our model achieves superior generation quality as indicated by the vertical axis in Figure 7. This indicates that the initial velocity field produced by our approach is better than the original rectified flow. We provide more qualitative results in Appendix H. Further results clarify the upper bounds of reflow. We provide the CIFAR-10 unconditional generation qualities of pre-trained diffusion models in Appendix J.

## 4.2 STRAIGHTNESS

We evaluate the straightness of the trajectories induced by the velocity field using the curvature, following prior works (Liu et al., 2022; Lee et al., 2023). Straightness is important because a straighter ODE trajectory causes fewer numerical errors due to discretization when using a small number of sampling steps, resulting in higher quality images (Stetter et al., 1973; Chen et al., 2023). For any continuously differentiable process $\mathbf{Z} = \{Z_t\}$, the curvature is measured by

$$S(\mathbf{Z}) = \int_0^1 \mathbb{E}\left[\left\|(Z_1 - Z_0) - \dot{Z}_t\right\|^2\right] dt \tag{15}$$

Additionally, it is known that $\mathbf{S}(\mathbf{Z}) = 0$ indicates exact straightness.

**Relationship between curvature and initial velocity delta (IVD)** Although a straight path often correlates with high-quality image generation in few steps, it is not a sufficient condition for effective 1-step sampling. This is because 1-step sampling depends entirely on the initially predicted velocity. Even with high curvature, an effective initial velocity can still guide the trajectory effectively toward the target space.

To address this limitation, we propose a new metric called Initial Velocity Delta (IVD), which directly measures the quality of 1-step generation by evaluating the accuracy of the initial velocity prediction. The calculation method for IVD is provided in the equation below:

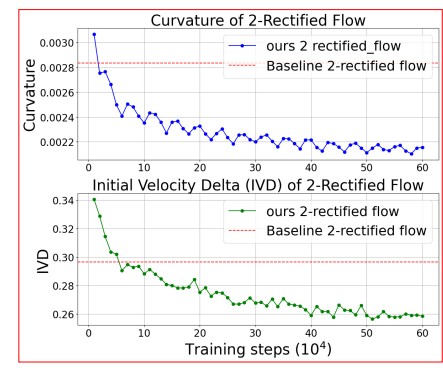

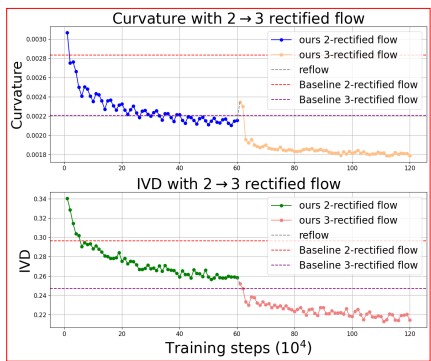

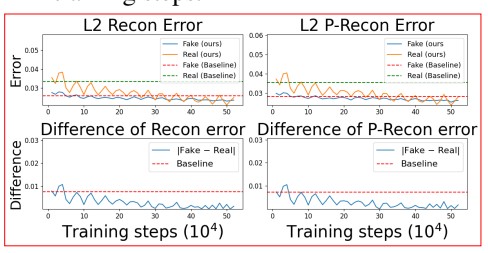

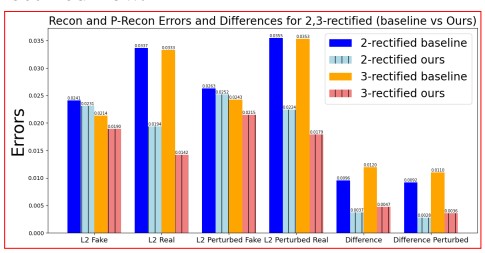

Figure 8: Curvature and IVD during training steps.

Figure 9: Reflow process from 2- to 3-rectified flow.

(a) Reconstruction and perturbed recon errors during training steps.

(b) Reconstruction, perturbed reconstruction errors, and their differences.

Figure 10: Comparison of reconstruction and perturbed reconstruction errors.

$$IVD(\mathbf{Z_t}, t_0) = \mathbb{E}\left[\left\|(Z_1 - Z_0) - \dot{Z}_{t_0}\right\|^2\right] \tag{16}$$

**Curvature and initial velocity delta** In this section, we compare the curvature and Initial Velocity Delta (IVD) between the original rectified flow and our balanced-conic rectified flow. Our approach demonstrates improved trajectory straightening and better preservation of the initial velocity direction. As shown in Figure 8, our method consistently achieves lower curvature and IVD values than the baseline, indicating a more stable trajectory even with fewer fake pairs. Furthermore, Figure 9 highlights that applying an additional reflow step from 2-rectified to 3-rectified flow further reduces both curvature and IVD, reinforcing the effectiveness of our training method.

### 4.3 RECONSTRUCTION WITH PERTURBATION TO ADDRESS DRIFT FROM THE REAL DISTRIBUTION

§ 3.1 showed that using only fake pairs in the reflow step leads to a disparity in reconstruction loss between real and fake images. This causes the model to overfit to fake images and disrupting smooth trajectories for real ones. Here, we experimentally demonstrate that incorporating real pairs in the reflow step effectively preserves real image trajectories compared to the original rectified flow.

As shown in Figure 10a, while the original rectified flow model exhibits a noticeable difference in reconstruction error between real and fake images, our method progressively reduces this difference with further training.

The results in Figure 10b can be interpreted from two perspectives: (1) Reconstruction Error: Our method better preserves the trajectory of real images and prevents overfitting to fake images compared to the baseline. (2) Perturbed Reconstruction Error: Lower perturbed reconstruction error suggests that our model better aligns the velocity field near real images, enhancing robustness to perturbations. For additional qualitative results, see Appendix G

### 4.4 FINE-TUNING WITH REAL PAIRS IMPROVES A PRE-TRAINED RECTIFIED FLOW MODEL

We demonstrate the effect of fine-tuning a pretrained rectified flow model using only 60,000 real pairs. For this experiment, we used the official rectified flow CIFAR-10 checkpoints available on GitHub.[5] With minimal additional training, we observe a noticeable improvement in 1-step quality

---

[5]Checkpoints are available at https://github.com/gnobitab/RectifiedFlow.

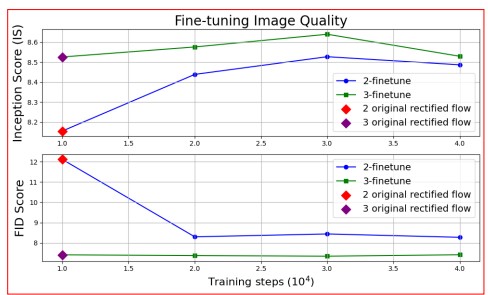 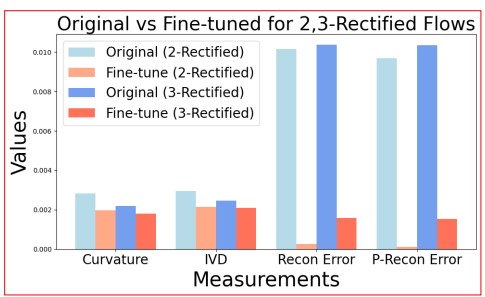

Figure 11: Comparison of image quality between the original rectified model and our fine-tuned model across training steps.

Figure 12: Comparing measurements for the original and fine-tuned models for 2- and 3-rectified flows.

as shown in Figure 11. Additionally, both curvature and IVD values decrease rapidly as illustrated in Figure 12. These results show that applying our method to a 2- or 3-rectified flow model, previously trained with standard techniques, effectively improves 1-step generation quality even with a small number of real pairs. Figure 12 also shows fine-tuned version has lower curvature and IVD than the original, indicating that our method is more straight than the original. Furthermore, the fine-tuned version has lower recon and p-recon differences between real and fake images, indicating that our method reduces the bias toward fake samples. For visualization convenience, IVD was scaled by $10^{-2}$.

### 4.5 HIGH-RESOLUTION IMAGE GENERATION

In this section, we assess the generalizability of our method on the LSUN Bedroom dataset (Yu et al., 2015) at a resolution of 256x256. We use the same hyperparameters, time schedule, and EMA settings as in the experiments by Liu et al. (2022).

We used 60K fake pairs and 5K real pairs, while the baseline used 120K fake pairs. Despite GPU limitations on the larger LSUN dataset, our method consistently outperformed the baseline in image quality. Figure 13 shows superior 1- and 2-step generation quality, and with adaptive step sampling (RK45), our approach achieved comparable quality with significantly fewer fake pairs than the baseline. Further details on the training configurations are provided in Appendix I.

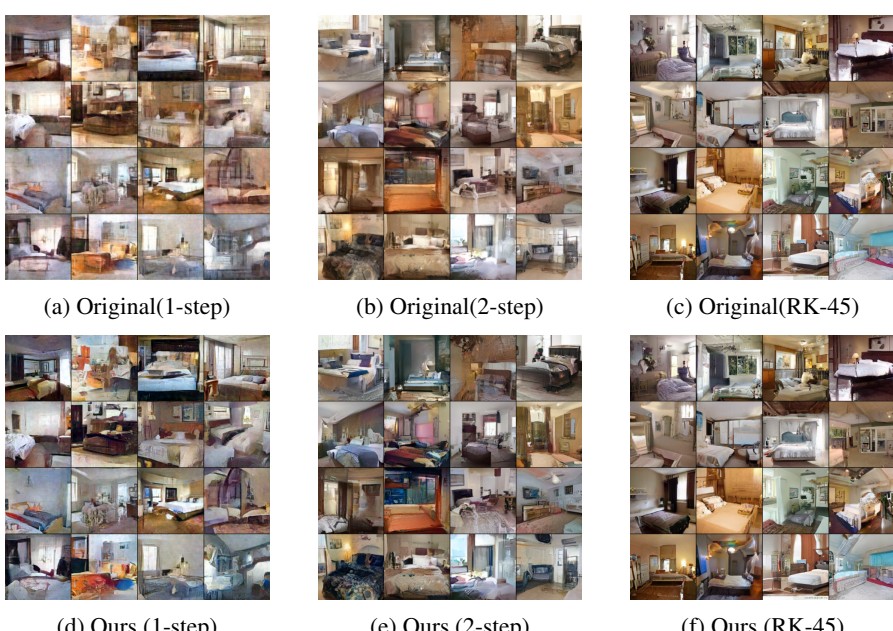

(a) Original(1-step)    (b) Original(2-step)    (c) Original(RK-45)

(d) Ours (1-step)    (e) Ours (2-step)    (f) Ours (RK-45)

Figure 13: Qualitative result of original and our 2-rectified flow results on the LSUN bedroom dataset.

## 4.6 ABLATION STUDY

We report results from an ablation study on various settings of our proposed framework, comparing four configurations: (1) without Slerp noise; and (2) reflow using just a single real pair; (3) our full method; and (4) the baseline 2-rectified flow. The comparison focuses on 1-step generation quality, curvature, IVD(Initial Velocity Delta), reconstruction error, and perturbed reconstruction error. For more detail, see Appendix A.

**Impact of incorporating real pairs in training** As shown in Table 2, configurations that include real pairs in training consistently outperform the baseline in terms of 1-step generation quality. This indicates that incorporating real pairs not only helps preserve the target distribution but also improves the trajectory influenced by lower-quality fake pairs.

**Benefits of adding noise via Slerp** Adding noise via Slerp avoids trajectory crossover, preserving a straighter path relative to real images. This leads to improved trajectory quality and enhances 1-step sampling efficiency. Moreover, using Slerp results in lower reconstruction error for real images compared to not using it. This demonstrates that adding noise through Slerp helps maintain the trajectory and neighborhood between reverse noise corresponding to real images more stably.

**Fixed Real Pair** This setting fixes the real pair just once and does not continuously update real pairs through repairing. Ours achieves better FID, IS, and lower reconstruction and p-reconstruction errors for real images than the single real pair setting. This suggests that periodically refreshing reverse noise during training helps preserve real image trajectories.

| Model | FID | IS | Curvature | IVD | Recon Error | | Perturbed Error | |
| --- | --- | --- | --- | --- | --- | --- | --- | --- |
| | | | | | Real | Fake | Real | Fake |
| Baseline | 12.21 | 8.08 | 0.002837 | 0.295078 | 0.033668 | 0.024106 | 0.035481 | 0.026270 |
| **Ours (slerp + conic)** | **5.98** | **8.79** | **0.002295** | 0.253334 | **0.019404** | 0.023139 | **0.022382** | 0.025206 |
| Fixed Real Pair | 6.69 | 8.59 | 0.002313 | 0.242444 | 0.020227 | 0.020607 | 0.022890 | 0.022914 |
| No slerp | 6.60 | 8.57 | 0.002322 | **0.240884** | 0.023380 | **0.020154** | 0.026063 | **0.022496** |

Table 2: An ablation table comparing various 2-rectified models across multiple metrics: FID, IS(1-step), Curvature, IVD, Recon and P-Recon errors.

## 5 LIMITATION AND DISCUSSION

Balanced conic rectified flow integrates real pairs into training by using Slerp to adjust noise of pairs towards the Gaussian manifold. This approach significantly improves the quality of 1-step generation compared to the baseline, even when using few fake pairs, and enhances pre-trained rectified models that utilize only real pairs.

However, limitations remain. First, while empirically effective, the practice of halving the amount of noise added by Slerp at each conic step lacks theoretical justification. Determining a better schedule for noise addition remains an open question. Second, since our method also uses fake pairs during the reflow process, it cannot completely resolve the slight degradation in many-step image generation quality in the reflow procedure. Another limitation is the extension of conic reflow—which adds noise—to Image-to-Image translation. A straightforward approach might be to train an image-to-noise-to-image model, but further investigation is needed in this area.

## 6 CONCLUSION

In conclusion, the balanced conic reflow method addresses key limitations of traditional rectified flow, which relies heavily on fake pairs and leads to a lower-quality velocity field during the re-flow process. By incorporating real pairs into training, even with a small number of fake pairs, our approach significantly improves the generation quality of k-rectified flow. Its simplicity and versatility make it applicable to various rectified flow-based generative models like SD3 and InstaFlow. Also, there is potential for improvement through techniques like applying additional loss functions or combining with diffusion-based fake pair generation (Lee et al., 2024).

We demonstrated empirically that our method forms a velocity field with lower curvature and initial velocity delta. This approach can positively impact not only image generation but also text-to-image/-video generation and editing tasks, offering improved quality and control in future work.

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

## A DETAILED SETTINGS

**Using exponential distribution**  As proposed in the work of Lee et al. (2024), the trajectory crossovers during the reflow process are more frequent near the noise or image endpoints (i.e., when closer to $X_0$ or $X_1$). To focus the training more effectively on these regions with high crossover frequency, we employed an exponential distribution. This phenomenon can be directly observed in Figure A1a, where we computed the top-k indices of the predicted velocity during the sampling process. The curvature is significantly higher near the start and end indices, indicating more crossovers at those points. Based on this observation, we adopted an exponential distribution rather than a uniform distribution, as illustrated inFigure A1b.

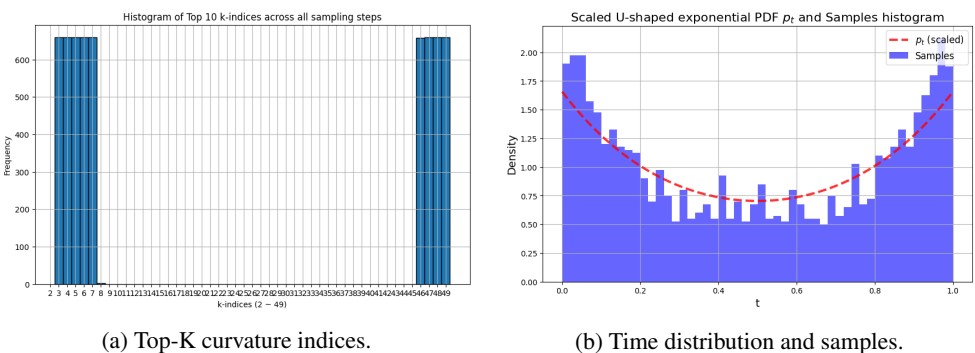

(a) Top-K curvature indices.  (b) Time distribution and samples.

Figure A1: Top-K (K=10) curvature indices and U-shape exponential time distribution with sampled data. For visualization convenience, the U-shaped probability density function $p_t$ was scaled so that the area under the curve equals 1.

$U_{\text{fake}}$ **and** $U_{\text{real}}$  The training schedule for conic reflow and original reflow is split into two phases. In the first half, conic and original reflows alternate to prevent bias toward either real or fake pairs. In the second half, only the original reflow is used to address the data imbalance, as fake pairs are far more abundant than real pairs. For instance, if total training step is $\mathbb{N} = 100$, then $U_{\text{real}} = \{1, 3, 5, 7, ..., 49\}$ and $U_{\text{fake}} = \{2, 4, 6, 8, ..., 50, 51, 52..., 100\}$.

**Experimental settings of Ablation study**  We conducted an ablation study with the following settings. All other settings remain consistent with those described in § 4 and Appendix I.

**No Slerp**: In this setting, we did not apply noise to the random sample $\epsilon \sim \mathcal{N}(0, I)$ using slerp, but rather used real and fake pairs to train according to our training scheme. The intention of this experiment was to observe the impact of using slerp for sampling $\epsilon$ in the conic reflow process.

## B SEVERAL SLERP NOISE PATTERNS AND LERP

In this section, we compare the 1-step generation quality of 2-rectified flow trained with various Slerp noise patterns and Lerp as a replacement for Slerp. SLERP noise patterns are explored including those that gradually increase from small noise and then decrease as well as strictly increasing and strictly decreasing patterns. The strictly increasing pattern allows us to analyze generation quality as noise progressively grows, while the strictly decreasing pattern helps us evaluate performance as noise is gradually removed. The comparison focuses on the performance of these settings in terms of Inception Score (IS) and Fréchet Inception Distance (FID).

All experiments are conducted with a batch size of 256, and training is performed for 300K iterations. Each setting uses 300K fake pairs and 60K real pairs for training. Other configurations remain identical to those described in § 4 and Appendix I.

**Noise scheduling patterns:**

- **Strictly increasing:** Noise increases from 0.006 to 0.13.

- **Strictly decreasing:** Noise decreases from 0.13 to 0.006.

- **Lerp:** Linear interpolation between $v^{-1}(X_1)$ and noise, defined as $(1 - \zeta)v^{-1}(X_1) + \zeta \cdot \epsilon$, where $\zeta \in [0, 0.13]$ (0.13 is the maximum noise level).

| Category | Method | NFE | IS ($\uparrow$) | FID ($\downarrow$) |
|---|---|---|---|---|
| **Slerp** | Ours | 1 | **8.72** | **6.63** |
| | Strictly Increasing | 1 | 8.48 | 6.64 |
| | Strictly Decreasing | 1 | 8.45 | 6.70 |
| **Lerp** | Linear Interpolation | 1 | 8.46 | 7.50 |

Table A1: Comparison of 1-step generation quality (IS, FID) for different Slerp noise scheduling patterns and Lerp. The highest values are bolded, and the second highest values are underlined.

As shown in Figure A1, we compare the impact of Slerp and Lerp noise scheduling patterns on 1-step generation quality in 2-rectified flow and observe the following:

1. **Slerp outperforms Lerp:** Slerp-based methods achieve better FID and IS compared to Lerp, suggesting that linear interpolation fails to capture the complex noise dynamics required for high-quality generation.

2. **Gradual noise increase is effective:** A strictly increasing noise schedule achieves better FID and IS than a strictly decreasing schedule, indicating that gradually introducing noise supports more stable training and improves generative quality.

3. **Ours vs Noise increasing:** While FID is nearly identical, Ours achieves higher IS, implying that our method generates images with greater diversity while maintaining comparable fidelity.

These results underscore the importance of carefully designing noise schedules to enhance both fidelity and diversity in generative tasks.

## C RECALL AND PRECISION

In this section, we evaluate the performance of our method compared to the original rectified flow using recall and precision metrics. These metrics allow us to analyze how well the generated data covers the real data distribution (recall) and how accurate the generated samples are compared to the real data (precision). (Kynkäänniemi et al., 2019)

We conduct the evaluation on the CIFAR-10 dataset, which consists of 60,000 real images combined with 50,000 synthetic images. For sampling, we utilize Euler sampling to ensure consistency across experiments.

As shown in Table A2, the results highlight key differences between multi-step and 1-step sampling:

- **Multi-Step Sampling**: Both our method and the original exhibit similar precision and recall values, indicating comparable performance in this setting.

- **1-Step Sampling**: Precision shows only a slight difference of approximately 0.85% on average, suggesting that both methods perform nearly identically in terms of precision. However, recall reveals a more significant advantage for our method:

  - On average, our method achieves 4.5% higher recall across all settings.
  - Specifically, for 2-Rectified Flow, our method outperforms the original by 5.5% in recall.

These findings indicate that while the precision of our method is nearly identical to the baseline, its higher recall demonstrates superior coverage of the real data distribution. Therefore, we can interpret these results as evidence that our method produces a more comprehensive and balanced representation of the underlying data distribution compared to the baseline.

| Method | NFE | Precision (↑) | Recall (↑) |
|--------|-----|---------------|------------|
| **2-Rectified Flow** | | | |
| *Full Step Generation* | | | |
| Ours | 104 | 0.691 | **0.605** (+0.005) |
| Original | 104 | **0.696** (+0.005) | 0.600 |
| *One step Generation* | | | |
| Ours | 1 | 0.687 | **0.583** (+0.055) |
| Original | 1 | **0.695** (+0.008) | 0.528 |
| **3-Rectified Flow** | | | |
| *Full Step Generation* | | | |
| Ours | 104 | 0.691 | **0.599** (+0.007) |
| Original | 104 | **0.698** (+0.007) | 0.592 |
| *One step Generation* | | | |
| Ours | 1 | 0.682 | **0.592** (+0.03) |
| Original | 1 | **0.691** (+0.009) | 0.562 |

Table A2: Comparison of 2- and 3-Rectified Flows under different NFE settings, highlighting the better results in bold.

## D  EXTREME NUMBER OF REFLOW PROCESS $(k = 4)$

In this section, we compare the generative quality, curvature, IVD of our method and the baseline under a setting with an extreme number of reflow processes ($k = 4$), which is higher than the typical settings of $k = 2$ or $k = 3$.

| Method | NFE | IS (↑) | FID (↓) | Precision (↑) | Recall (↑) |
|--------|-----|--------|---------|---------------|------------|
| **4-Rectified Flow** | | | | | |
| Ours | 100 | **9.076** | **4.195** | 0.696 | **0.585** |
| Original | 100 | 8.951 | 4.490 | **0.705** | 0.584 |
| Ours | 1 | **8.808** | **5.662** | **0.690** | **0.581** |
| Original | 1 | 8.597 | 6.580 | 0.688 | 0.576 |

| Method | Curvature (↓) | IVD (↓) |
|--------|---------------|---------|
| **4-Rectified Flow** | | |
| Ours | **0.00176** | **0.20787** |
| Original | 0.00186 | 0.21812 |

Table A3: Comparison of NFE, IS, FID, precision, and recall for 4-Rectified Flow between Ours and Original. The better values are highlighted in bold.

Table A4: Comparison of curvature and IVD for 4-Rectified Flow between Ours and Original rectified flow. The better values are highlighted in bold.

As shown in Table A3 and Table A4:

- **1-Step Generation Quality**: Our method outperforms the baseline in both FID and IS, demonstrating superior generative performance.
- **Adaptive Step Generation Quality**: Similarly, our method shows better results compared to the baseline.
- **Additional Metrics**: Our method achieves better curvature and IVD compared to the baseline. This indicates that our method forms a velocity field that enables the solution trajectory to be straighter during the reflow process and better preserves the direction of the initial velocity, ensuring it aligns more closely with the overall trajectory.

These findings show that our method preserves the distribution for real images while preventing bias toward fake images. Even with $k > 3$, our method improves the reflow process and makes it robust for extreme reflow step settings.

## E  USING EVEN LESS FAKE SAMPLES

**Extreme case (60k fake pair(baseline) and 60k + kernel pair(ours))**  Figure A2 compares the generation quality of the 2-rectified flow after reflow, using an extremely small number of fake pairs. As shown in the figure, consistent with the observations by Liu et al. (2022), when the number

of fake pairs is too small, the 1-step image generation quality slightly improves. However, due to excessive saturation during reflow training with a limited fake image distribution, the many-step image generation quality deteriorates despite the improvement in 1-step quality.

In contrast, our approach preserves the path for real images during the conic reflow step, even with an extremely small number of fake pairs (60k). This results in not only improved 1-step quality but also mitigates the degradation in many-step generation quality compared to the baseline.

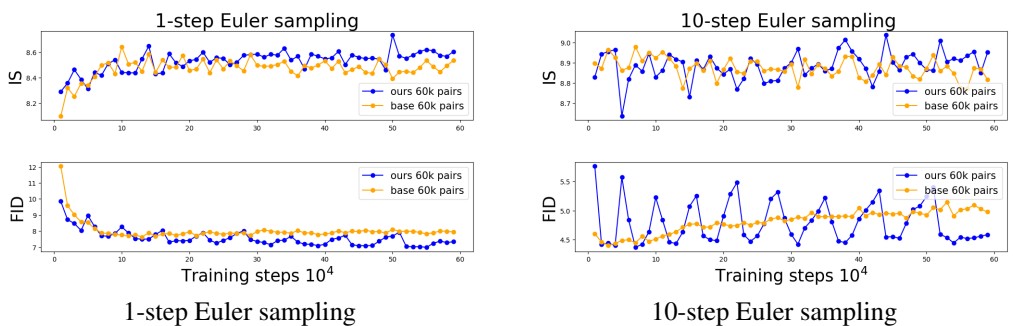

|        1-step Euler sampling        |        10-step Euler sampling        |

Figure A2: FID and IS score for baseline and ours 2-rectified flow (using 60k fake pairs)

## F  GENERALIZATION TO HIGH-RESOLUTION: LSUN BEDROOM

Table A5 and Figure A3 compares the original rectified flow and ours in generating high resolution images (LSUN bedroom). We report IVD, curvature, recon error, perturbed recon error for fake and real images. Furthermore, Figure A4, A5, A6, A7, A8, A9, A10, A11 provide qualitative comparison on LSUN bedroom.

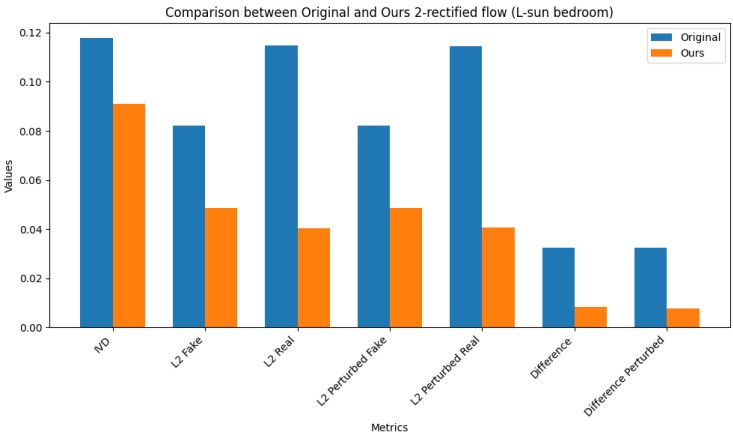

Figure A3: Comparison plot for Original and Ours 2-rectified flow (Lsun Bedroom)

|          | IVD    | L2 Fake | L2 Real | L2 Perturbed Fake | L2 Perturbed Real | Difference | Difference Perturbed |
|----------|--------|---------|---------|-------------------|-------------------|------------|----------------------|
| Original | 1.1790 | 0.0822  | 0.1147  | 0.0820            | 0.1146            | 0.0325     | 0.0326               |
| Ours     | 0.9103 | 0.0487  | 0.0405  | 0.0486            | 0.0407            | 0.0083     | 0.0079               |

Table A5: Comparison of metrics for Original and Ours 2-rectified flow (Lsun Bedroom)

# G  QUANTITATIVE RESULTS OF RECONSTRUCTION AND PERTURBED RECONSTRUCTION

Figure A12, A13, A14, A15 provide quantitative comparison of reconstruction between the baseline and ours. Figure A16, A17 provide quantitative comparison of *perturbed* reconstruction between the baseline and ours.

# H  IMAGE QUALITY WITH DIFFERENT NUMBER OF EULER STEPS

Figure A18, A19, A20, A21, A22, A23, A24, A25 provides qualitative results on CIFAR10 with 1,2, and 100 Euler steps.

# I  MISC. CONFIGURATIONS

Our experiments on CIFAR10 is configured as follows. For rectified flow, we utilize the same network architecture as the rectified flow model based on DDPM++ in (Song et al., 2020b; Liu et al., 2022) with batch size 256. The training process is smoothed using an exponential moving average (EMA) with a decay rate of 0.999999, following the approach in Song et al. (2020b). The Adam optimizer (Diederik, 2014) is used with a learning rate of 2e-4, and a dropout rate of 0.15 is applied.

For the high-resolution (LSUN), both our method and the baseline 2-rectified flow were trained for 300,000 steps with a batch size of 16. Training started from the same 1-rectified flow checkpoints from the official rectified flow repository. During inference, all models used fixed seeds and identical noise conditions corresponding to the sampling steps.

# J  UNCONDITIONAL GENERATION MODEL QUALITY

| Method | NFE (↓) | IS (↑) | FID (↓) |
|---|---|---|---|
| ***Full Simulation (Euler Solver, N=2000)*** | | | |
| VP SDE (Song et al., 2020b) | 2000 | 9.58 | 2.55 |
| sub-VP SDE (Song et al., 2020b) | 2000 | 9.56 | 2.61 |
| NCSN++ (VE SDE) (Song et al., 2020b) | 2000 | 9.83 | 2.31 |
| DDPM (Ho et al., 2020) | 1000 | 9.46 | 3.21 |
| ***Adaptive Step Simulation (Runge–Kutta (RK45), Adaptive N)*** | | | |
| VP ODE (Song et al., 2020b) | 140 | 9.37 | 3.93 |
| sub-VP ODE (Song et al., 2020b) | 146 | 9.46 | 3.16 |
| NCSN++ (VE ODE) (Song et al., 2020b) | 176 | 9.35 | 5.38 |
| LSGM (Vahdat et al., 2021) | 147 | - | 2.10 |
| PFGM (Xu et al., 2022) | 110 | 9.68 | 2.35 |
| EDM (Karras et al., 2022) | 35 | **9.84** | **2.04** |
| 1-Rectified Flow | 127 | 9.60 | 2.58 |
| 2-Rectified Flow | 110 | 9.24 | 3.36 |
| **2-Rectified Flow Ours** | 104 | 9.30 | 3.24 |
| 3-Rectified Flow | 104 | 9.01 | 3.96 |
| **3-Rectified Flow Ours** | 98 | 9.14 | 3.70 |
| ***One-Step Simulation (Euler Solver, N=1)*** | | | |
| VP ODE (+*Distill*) (Song et al., 2020b) | 1 | 1.20 (8.73) | 451 (16.23) |
| sub-VP ODE (+*Distill*) (Song et al., 2020b) | 1 | 1.21 (8.80) | 451 (14.32) |
| NCSN++ (VE ODE) (+*Distill*) (Song et al., 2020b) | 1 | 1.18 (2.57) | 461 (254) |
| 1-Rectified Flow (+*Distill*) | 1 | 1.13 (9.08) | 378 (6.18) |
| 2-Rectified Flow (+*Distill*) | 1 | 8.08 (9.01) | 12.21 (4.85) |
| **2-Rectified Flow Ours (+*Distill*)** | 1 | 8.79 (9.11) | 5.98 (4.16) |
| 3-Rectified Flow (+*Distill*) | 1 | 8.47 (8.79) | 8.15 (5.21) |
| **3-Rectified Flow Ours (+*Distill*)** | 1 | 8.84 (8.96) | 5.48 (4.68) |
| ***Diffusion + Distillation*** | | | |
| DDIM Distillation (Luhman & Luhman, 2021) | 1 | 8.36 | 9.36 |
| DMD (Yin et al., 2024) | 1 | - | 3.77 |
| Diff-Instruct (Luo et al., 2024) | 1 | 9.89 | 4.53 |
| PD (Salimans & Ho, 2022) | 1 | 8.69 | 8.34 |
| DFNO (Zheng et al., 2023) | 1 | (-) | 4.15 |
| SID, ($\alpha = 1.2$) (Zhou et al., 2024) | 1 | **9.98** | **1.92** |
| ***Consistency Model*** | | | |
| CD (Song et al., 2023) | 1 | 9.48 | 3.55 |
| CT (Song et al., 2023) | 1 | 8.49 | 8.70 |
| ICT (Song & Dhariwal, 2023) | 1 | 9.54 | 2.83 |
| CTM (Kim et al., 2023) | 1 | - | 5.19 |
| CTM + GAN (Kim et al., 2023) | 1 | - | 1.98 |

Table A6: Unconditional generation quality with various diffusion-based models on CIFAR-10. Blue rows highlight the top-5 baselines for 1-NFE, and red rows for Adaptive NFE (RK-45). The lowest FID and highest IS scores in each setting are bolded.

Figure A4: Original(up) and ours(down) 2-rectified flow(1-step, seed 1)

Figure A5: Original(up) and ours(down) 2-rectified flow(1-step, seed 2)

Figure A6: Original(up) and ours(down) 2-rectified flow(1-step, seed 3)

Figure A7: Original(up) and ours(down) 2-rectified flow(1-step, seed 333)

Figure A8: Original(up) and ours(down) 2-rectified flow(1-step, seed 555)

Figure A9: Original(up) and ours(down) 2-rectified flow(2-step, seed 785)

Figure A10: Original(up) and ours(down) 2-rectified flow(2-step, seed 9913)

Figure A11: Original(up) and ours(down) 2-rectified flow(164-step, seed 3125)

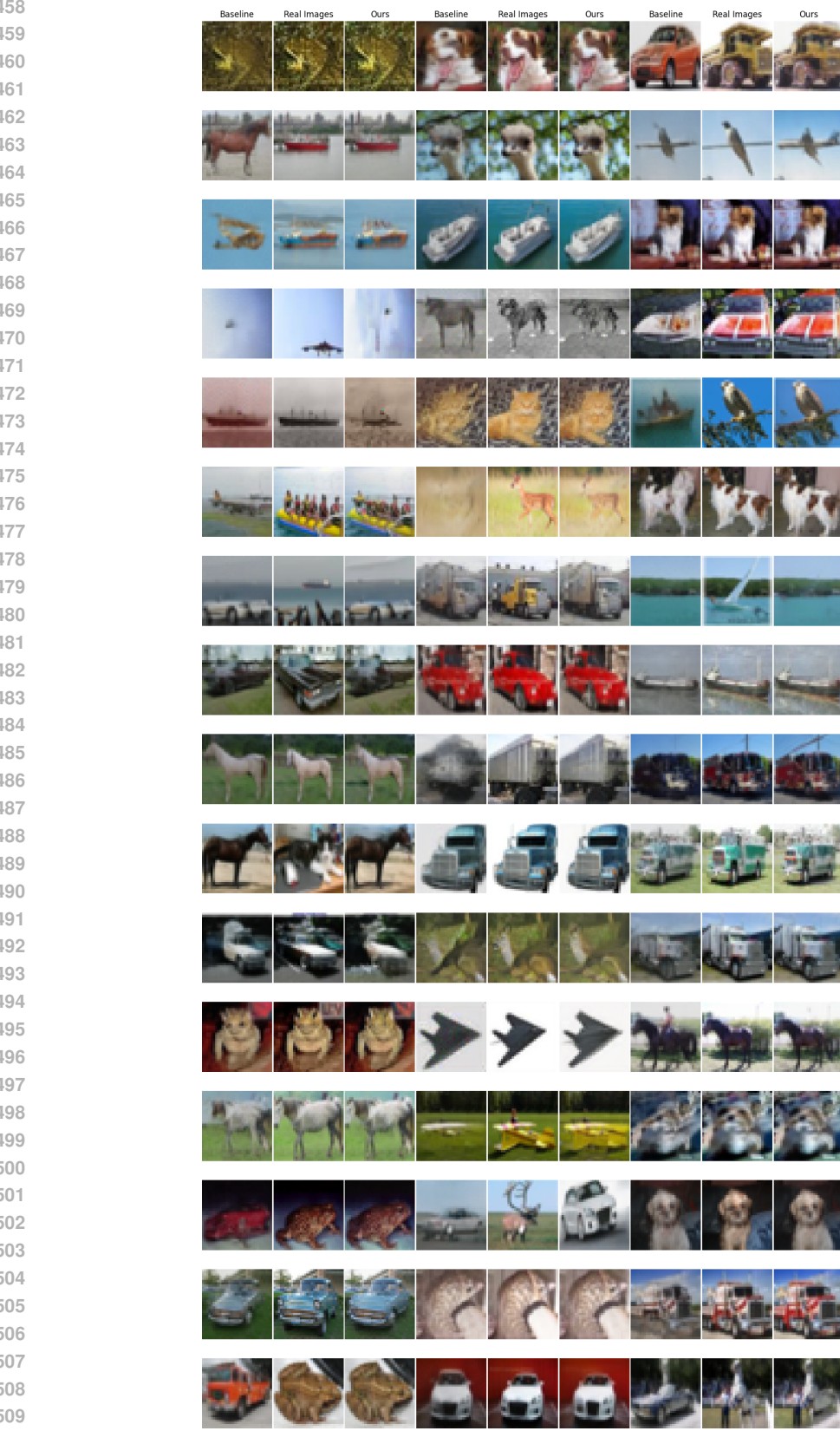

Figure A12: Compare with reconstruction image to baseline(1-step euler)

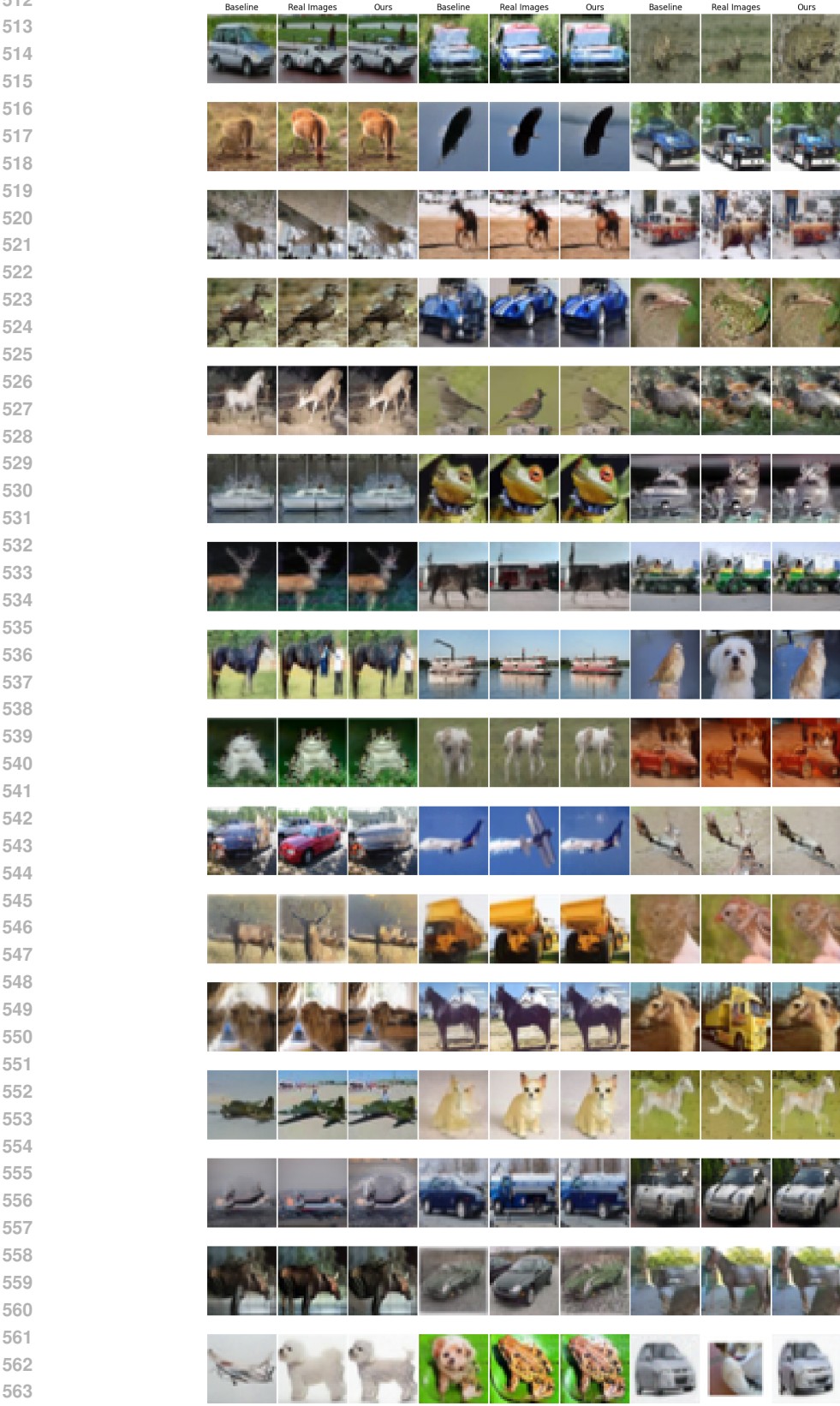

Figure A13: Compare with reconstruction image to baseline(1-step euler)

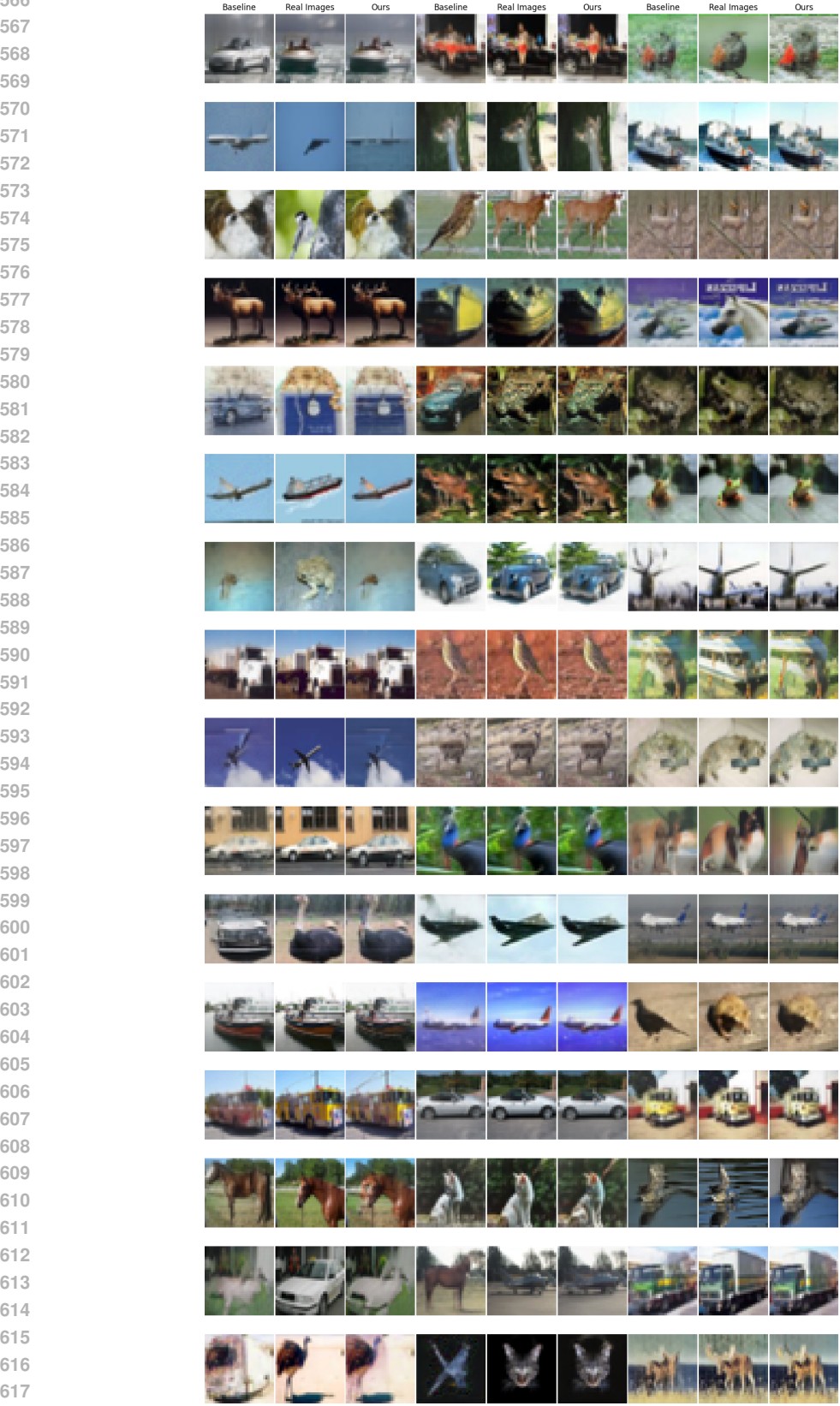

Figure A14: Compare with reconstruction image to baseline(1-step euler)

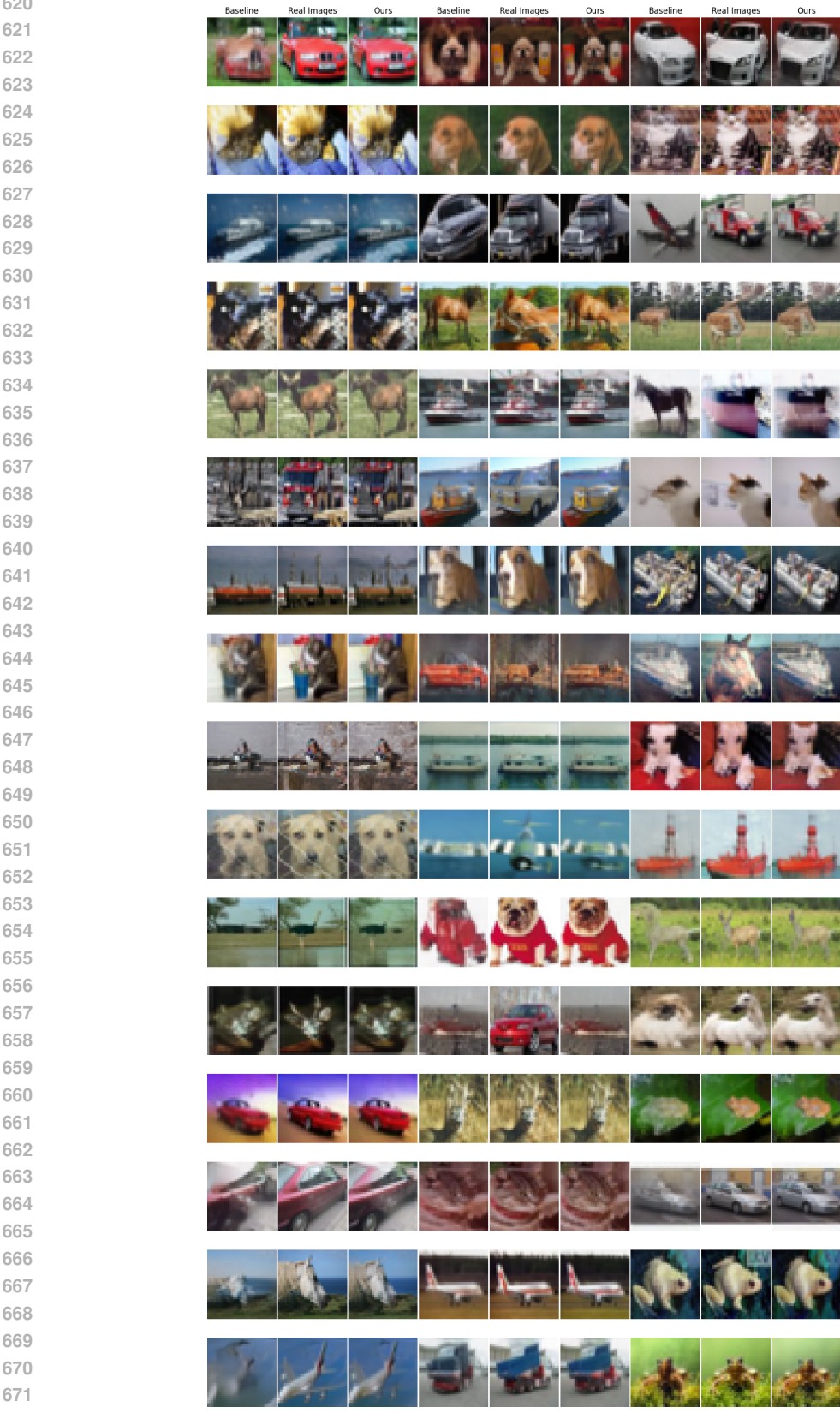

Figure A15: Compare with reconstruction image to baseline(1-step euler)

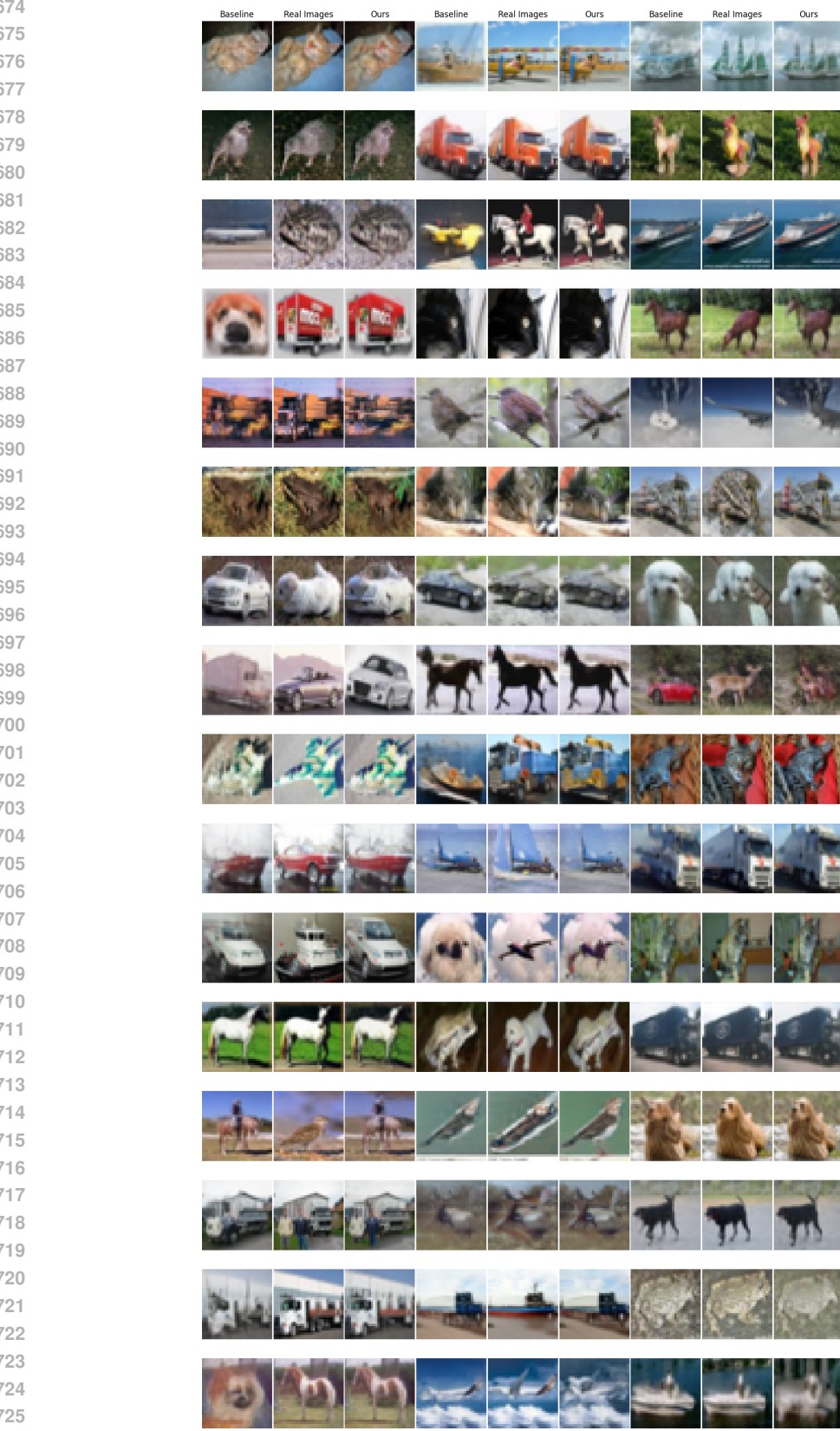

Figure A16: Compare with perturbed($\varepsilon = 0.04$) reconstruction image to baseline(1-step euler)

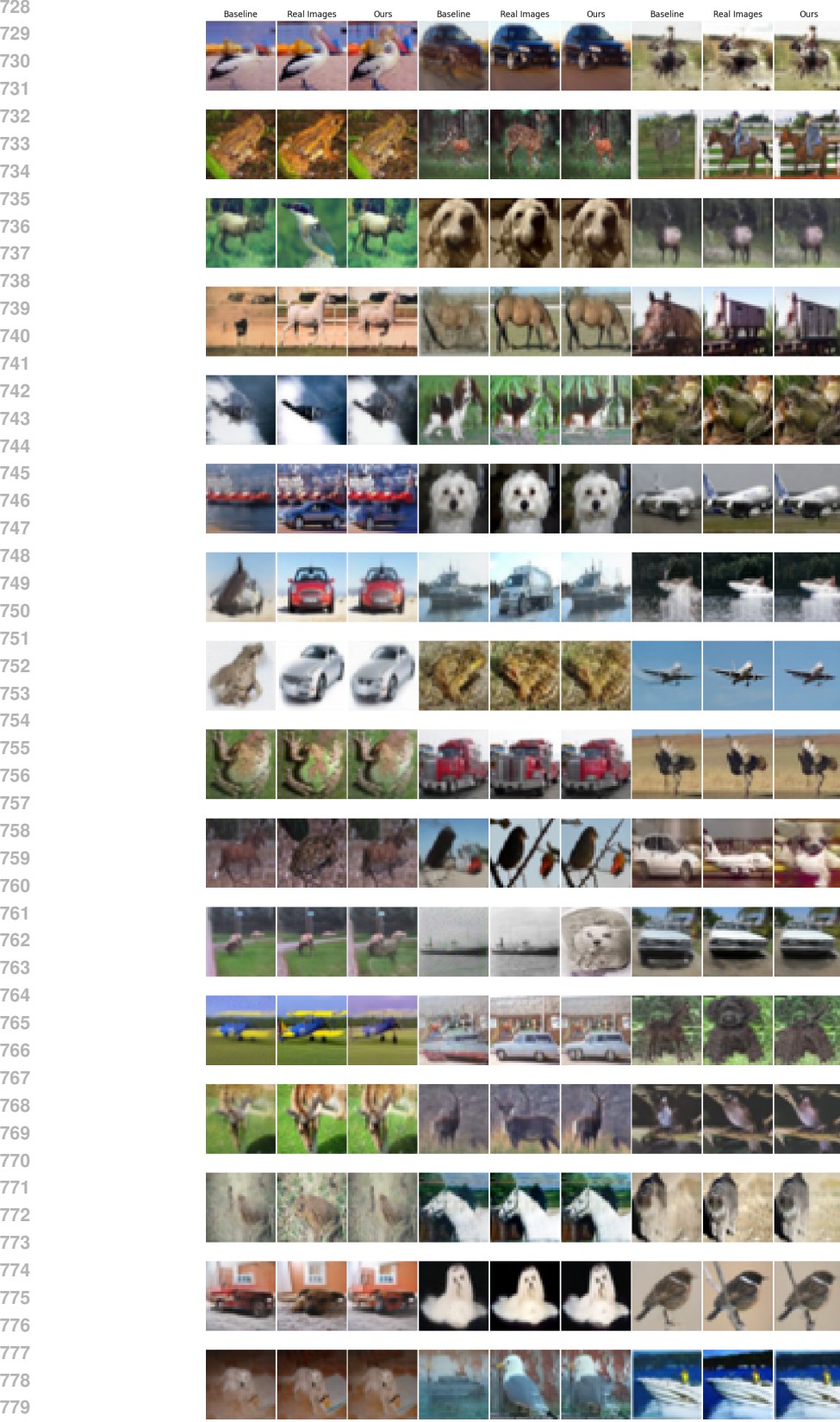

Figure A17: Compare with perturbed($\varepsilon = 0.04$) reconstruction image to baseline(1-step euler)

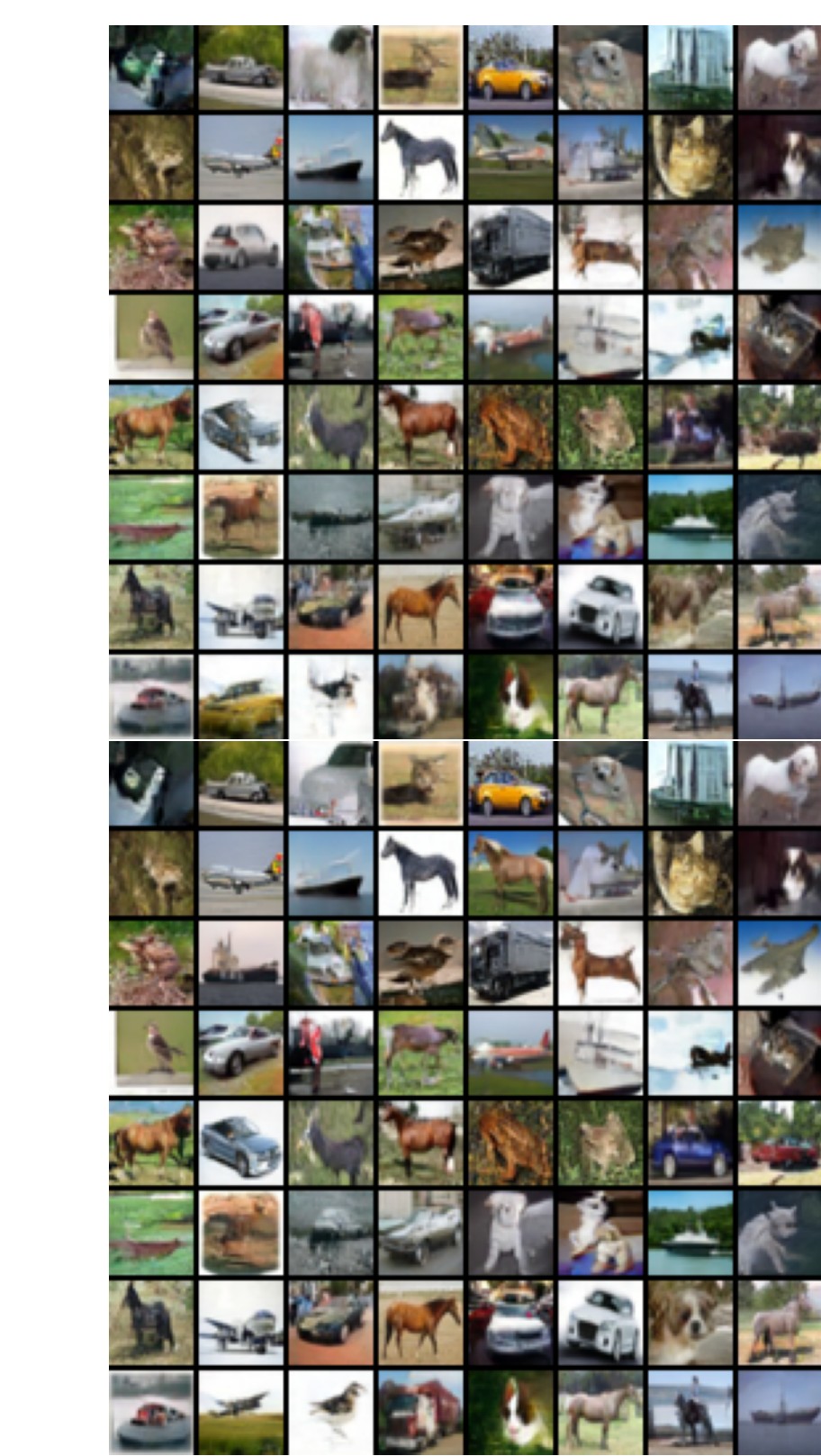

Figure A18: Original(up) and ours(down) 2-rectified flow(1-step, seed 1234)

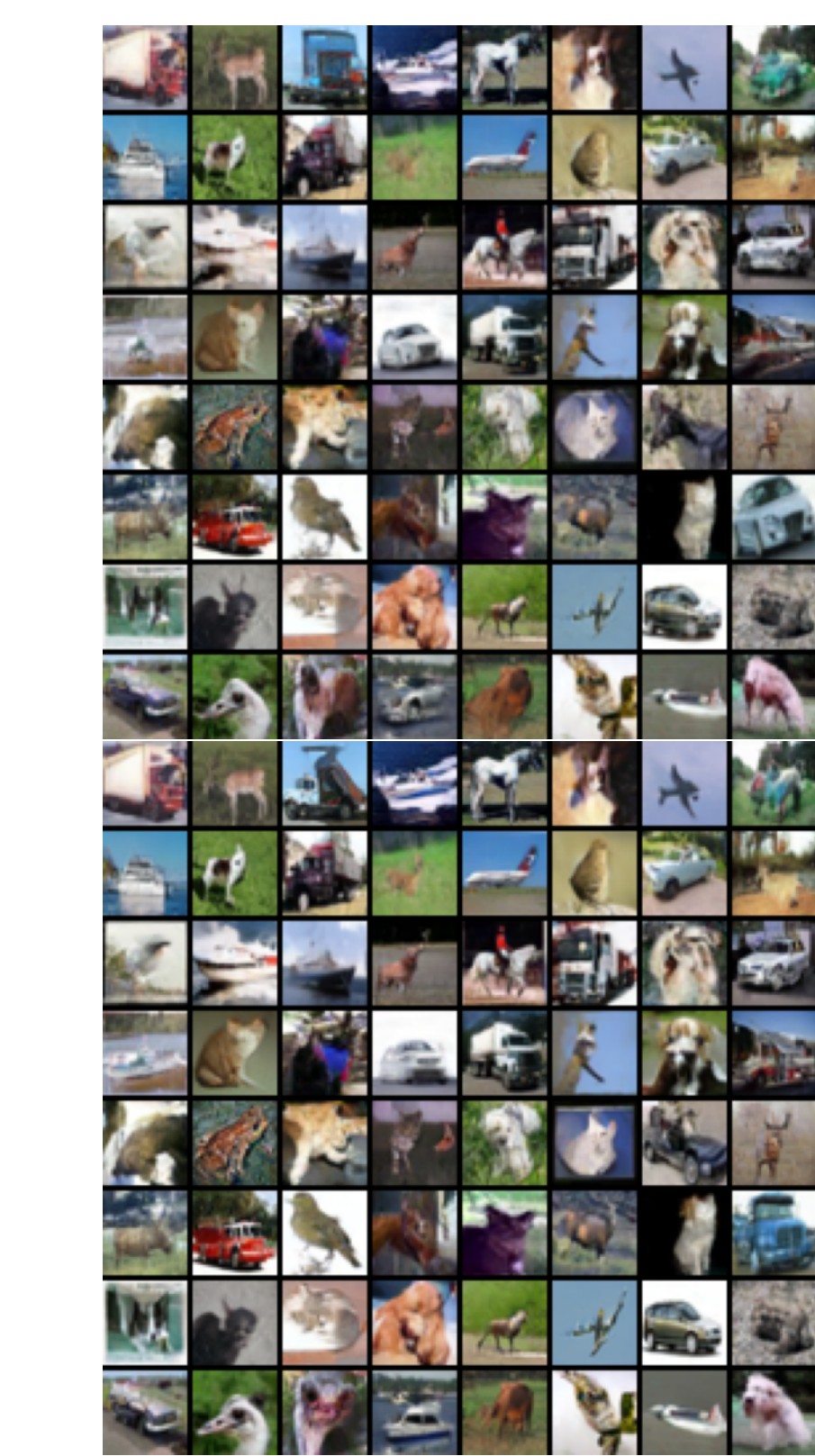

Figure A19: Original(up) and ours(down) 2-rectified flow(1-step, seed 1234)

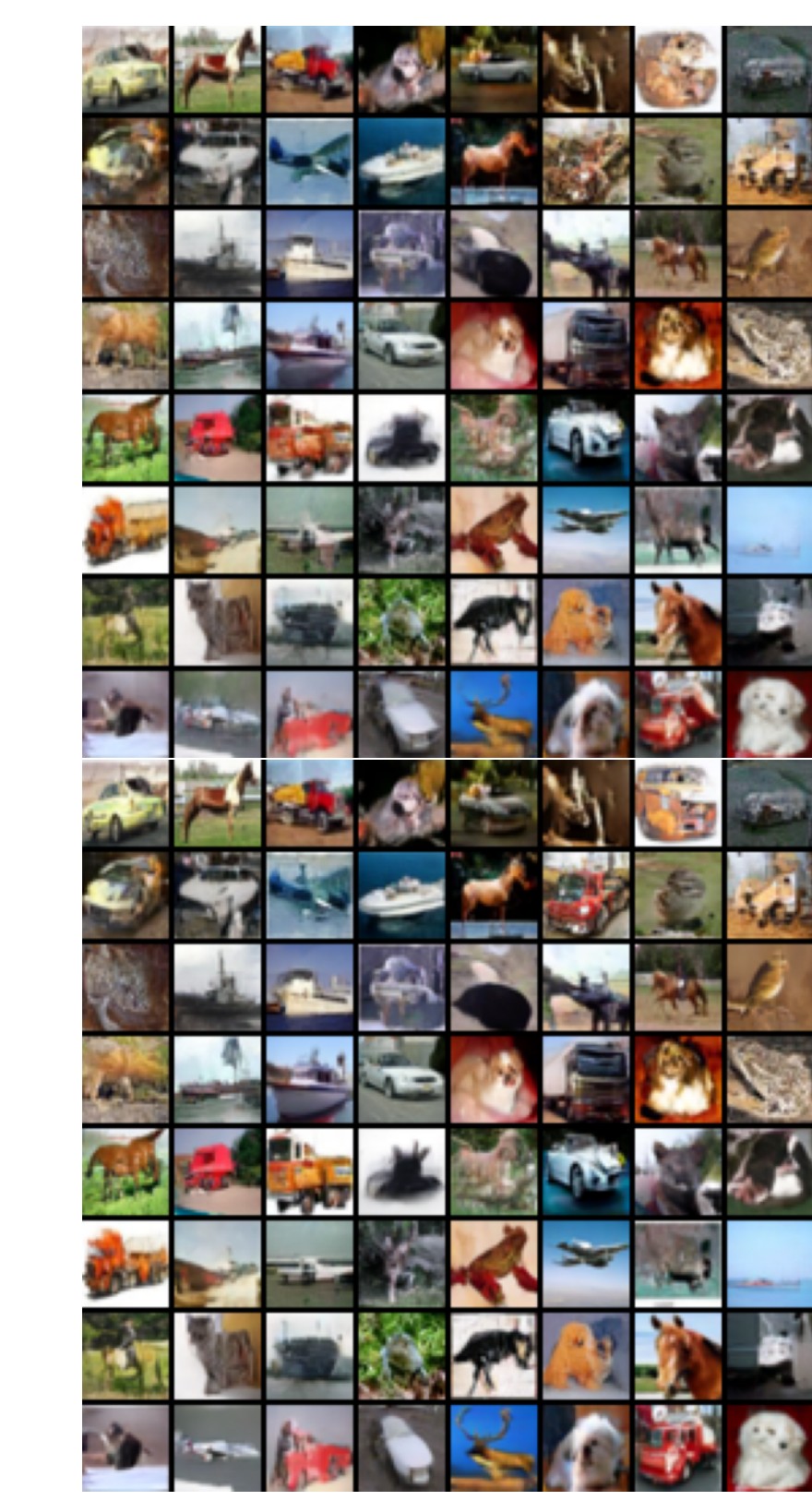

Figure A20: Original(up) and ours(down) 2-rectified flow(1-step, seed 1234)

1944
1945
1946
1947
1948
1949
1950
1951
1952
1953
1954
1955
1956
1957
1958
1959
1960
1961
1962
1963
1964
1965
1966
1967
1968
1969
1970
1971
1972
1973
1974
1975
1976
1977
1978
1979
1980
1981
1982
1983
1984
1985
1986
1987
1988
1989
1990
1991
1992
1993
1994
1995
1996
1997

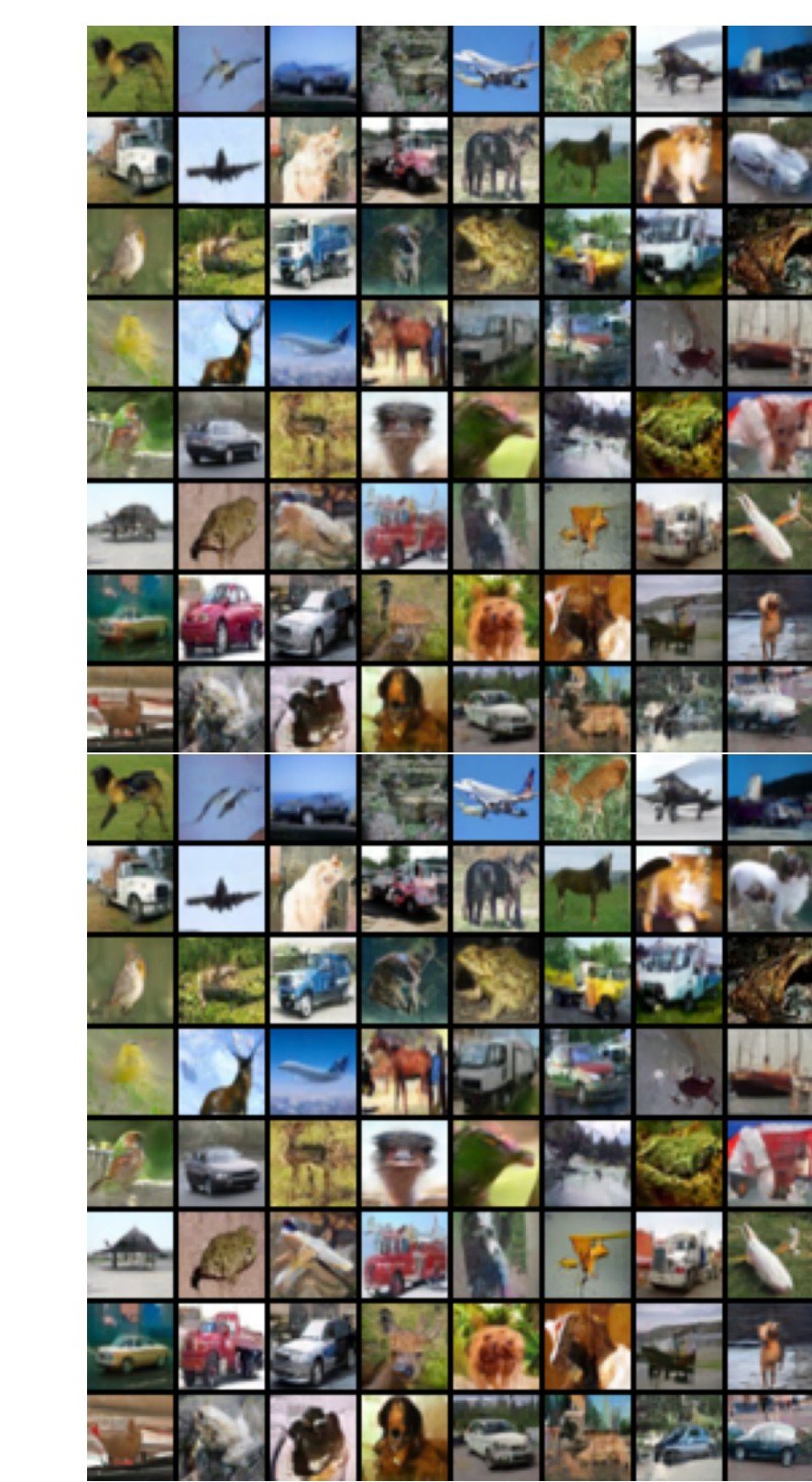

Figure A21: Original(up) and ours(down) 2-rectified flow(1-step, seed 1234)

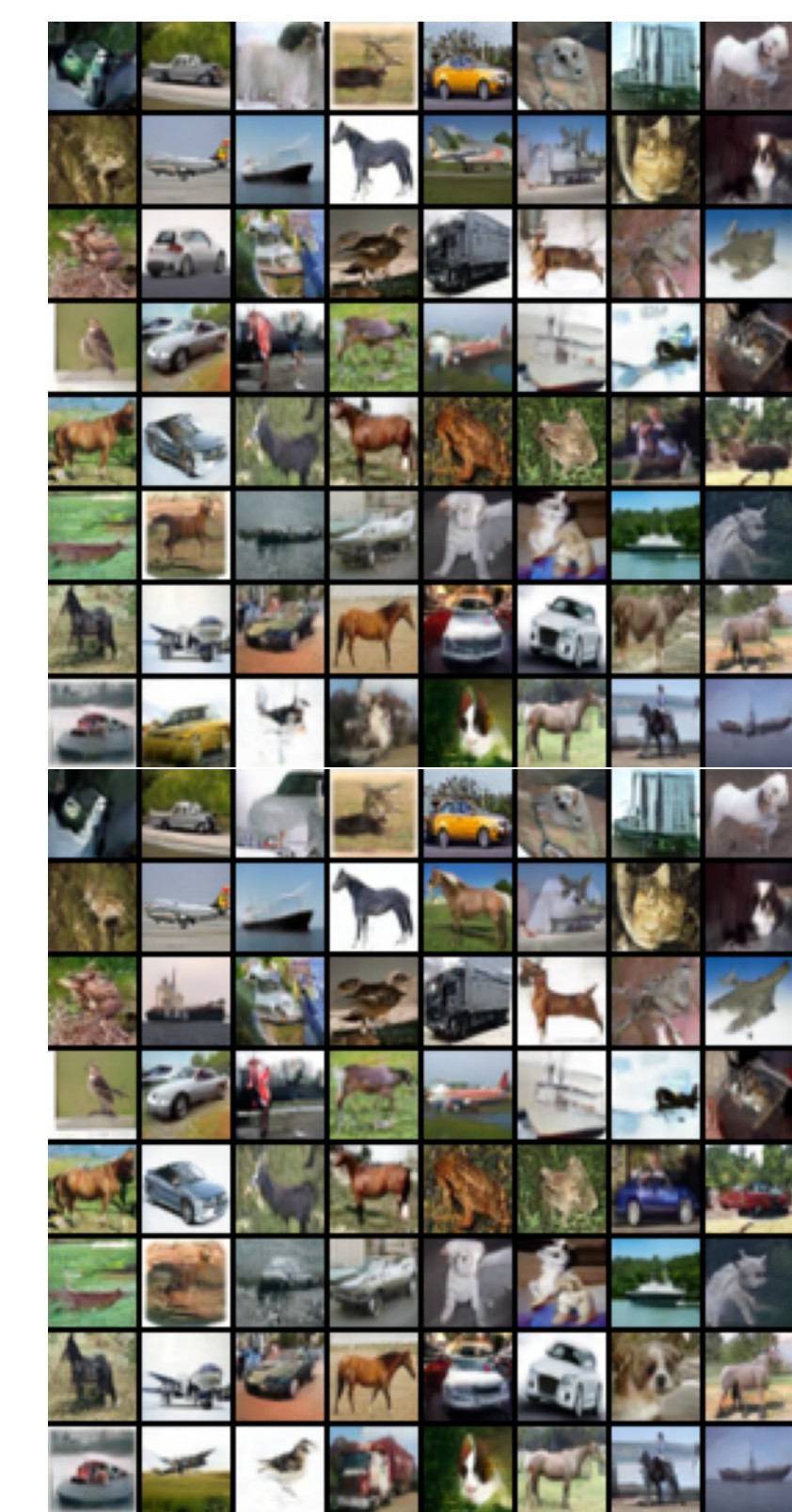

Figure A22: Original(up) and ours(down) 2-rectified flow(2-step, seed 1993)

2052
2053
2054
2055
2056
2057
2058
2059
2060
2061
2062
2063
2064
2065
2066
2067
2068
2069
2070
2071
2072
2073
2074
2075
2076
2077
2078
2079
2080
2081
2082
2083
2084
2085
2086
2087
2088
2089
2090
2091
2092
2093
2094
2095
2096
2097
2098
2099
2100
2101
2102
2103
2104
2105

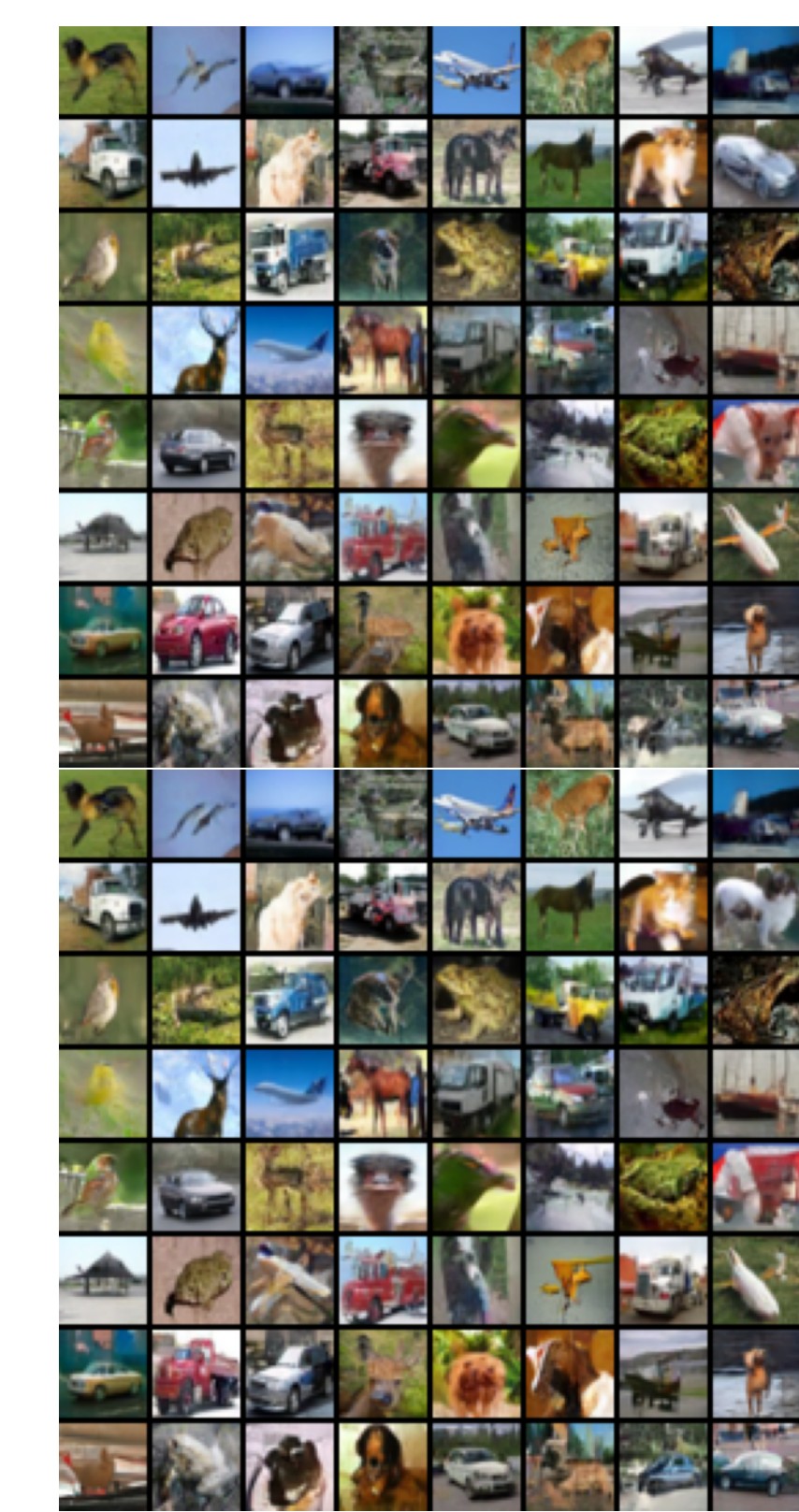

Figure A23: Original(up) and ours(down) 2-rectified flow(2-step, seed 1993)

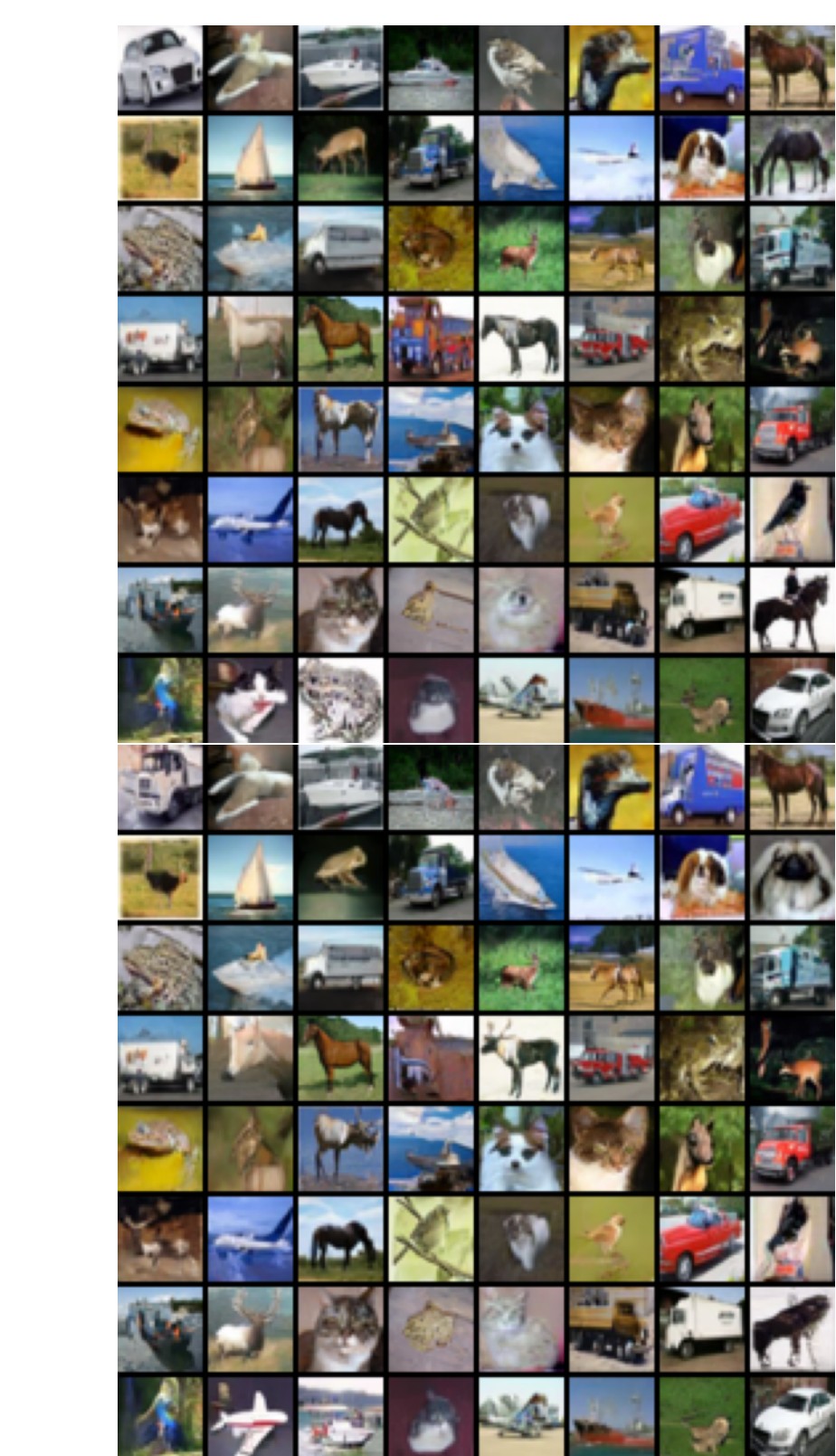

Figure A24: Original(up) and ours(down) 2-rectified flow(104-step, seed 2000)

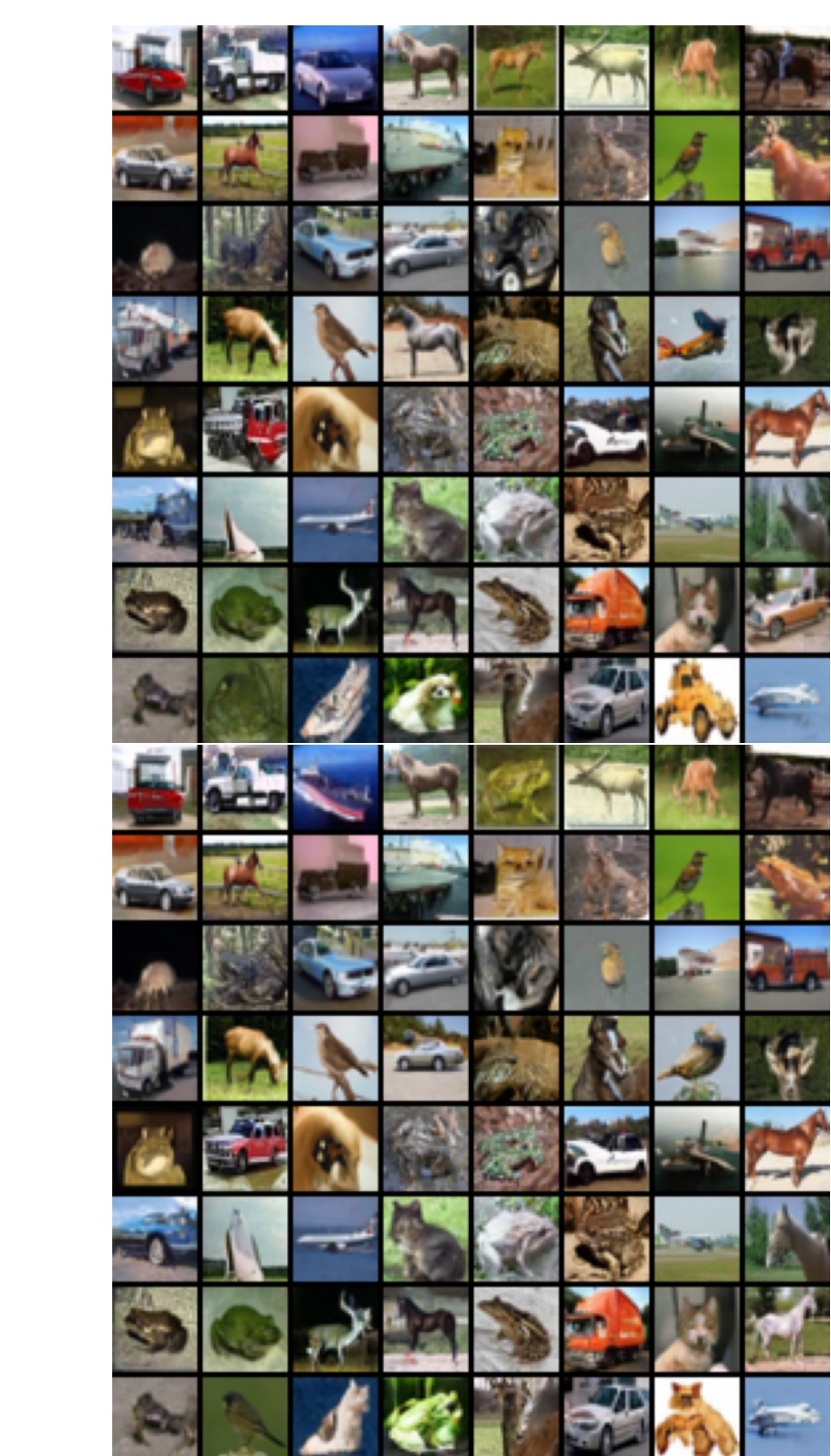

Figure A25: Original(up) and ours(down) 2-rectified flow(104-step, seed 2000)

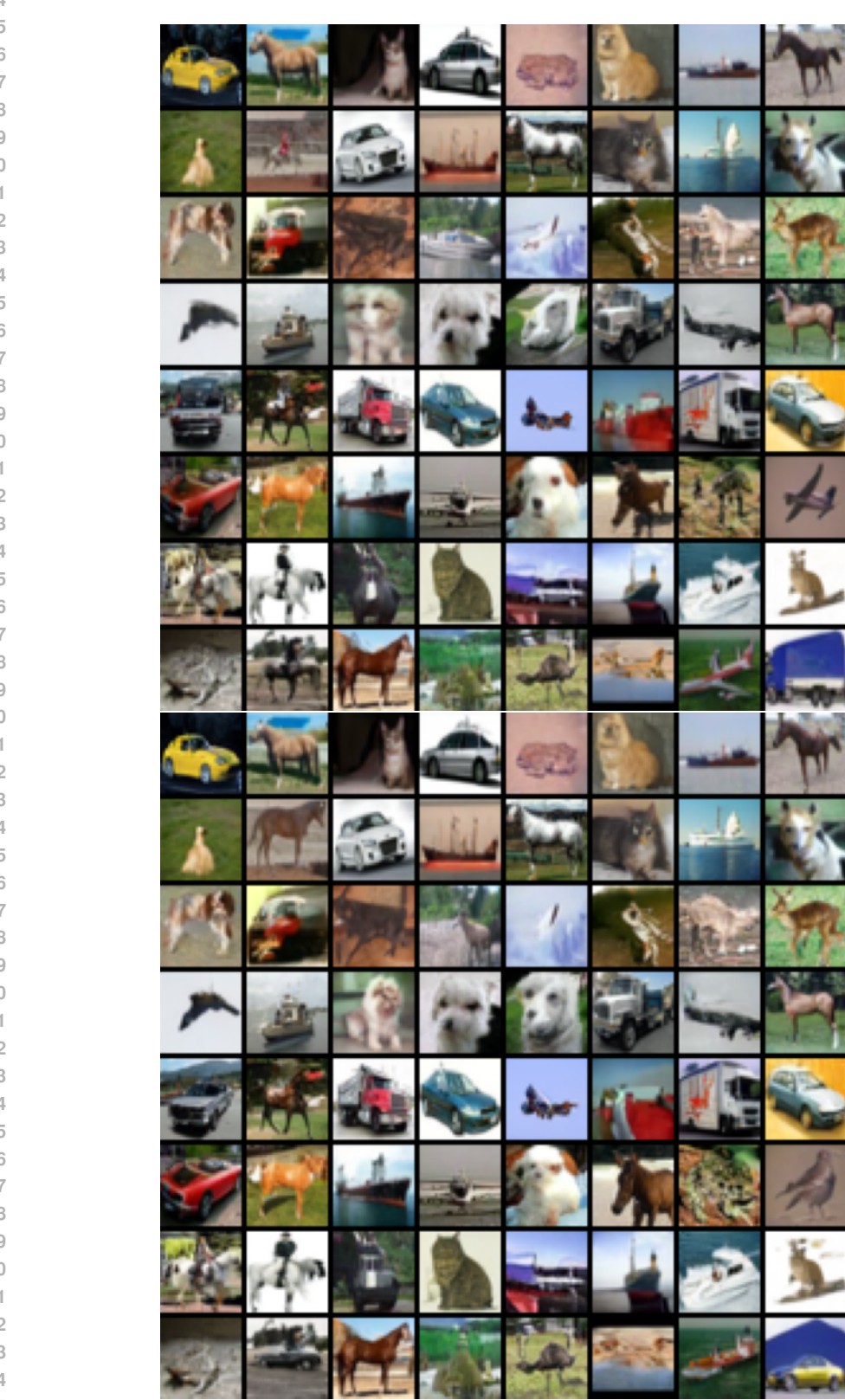

Figure A26: Original(up) and ours(down) 2-rectified flow(104-step, seed 2000)

