# OpenReview forum: "Balanced conic rectified flow"
_ICLR.cc/2025/Conference — Submitted to ICLR 2025_

### Official Review · Reviewer_et3P · 2024-10-15

**Soundness:** 3
**Presentation:** 3
**Contribution:** 2
**Rating:** 6
**Confidence:** 3

**Summary:**

This paper proposes to introduce real samples and their inverted noises (real sample -> noise) instead of commonly used Gaussian noises and generated samples (noise -> generated samples) for rectification (reflow). This operation alleviates the target distribution drift and improve the performance and efficiency of rectification.

**Strengths:**

Overall I think the paper is good and reading this paper inspire me to some extent,
1. The motivation is clear (as indicated in the abstracts and introductions) and the overall storytelling is smooth for me.
2. The demonstration is good (good figures and tables).
3. The method is straightforward and easy to understand and implement.
4. The performance seems to improve a lot compared to the baseline method.

**Weaknesses:**

1. **Performance**: Although the improvements from vanilla rectified flow, the performance still seems to be  not competitive with advanced distillation and acceleration techniques like iCT [1] and  SiD [2]. For example, in table 1, we can see that the proposed method with rf-3 achieves FID 5.48 (4.68), while in iCT achieves 1-step FID 2.83, 2.51 on CIFAR-10.

2. **Comparison**: For all tables, I would like the authors to add the performance of the original pretrained diffusion models. This will give the readers a clear understanding what is the upper-bound of reflow. Otherwise, readers might find it struggle to understand gap between a pretrained diffusion model  and a trained rectified model.

3. **Metrics**: I would like to see more metrics like recall and precision to better understand the gap of real data distribution and the generated data distribution.

4. **Typos**:  In line 746-747,  there is a `??` ; From line 753 to line 768, the words are horizontally centred in a strange way.


5. **Theoretical Analysis**: Though soundness, no critical theoretical analysis or bounds are provided. I'm not implying that there necessarily is one, but I am suggesting that adopting a rational analytical perspective can significantly enrich the substance of the article.

[1] Improved Techniques for Training Consistency Models. ICLR 2024

[2] Score identity Distillation: Exponentially Fast Distillation of Pretrained Diffusion Models for One-Step Generation. ICML 2024

**Questions:**

If the pretrained diffusion model will generate a data samples whose distribution has a drift from real data distribution, it will probably fail to reconstruct the noise with real  samples? That is to say, you can never really achieve a perfect coupling noise and sample with a spoiled pretrained diffusion model. How to handle these situations?

---

> ### Author Response · Authors · 2024-11-21
> **Response to Reviewer et3P**
>
> Thank you for taking an interest in our work and for your detailed questions.
>
> $\textbf{[W1,2] Performance and Comparison}$\
> Our study focuses on experimentally addressing the instability of the reflow process and proposing a method to mitigate it. As such, we primarily compared with the original rectified flow, as it is more relevant and better aligned with the objectives of our work, rather than other diffusion-based generative models.
>
> $\textbf{[W 3] Metrics}$\
> Thank you for the interesting suggestion. Calculating recall and precision can show how well our method balances diversity and fidelity compared to the baseline and evaluate the use of real pairs in reflow. We will include these metrics in the Appendix by November 25.
>
> $\textbf{[W 4] Typos}$\
> Thank you for pointing this out. We have corrected these issues.\
> $\bullet$ In line 746-747 ??  => Fig A12 ,... , A25.\
> $\bullet$ From line 753 to line 768 => Aligned with other pages.
>
> $\textbf{[W 5] Theoretical Analysis}$\
> Thank you for your valuable comment. Our approach focused on addressing numerical errors in the reflow process through experimental observations, making theoretical analysis beyond our scope. As noted in the Limitations, this is a valuable direction for future research.
>
>
>
> We hope these explanations resolve the concerns. Further questions or suggestions would be also appreciated.

---

> > ### Comment · Reviewer_et3P · 2024-11-25
> > **Reply**
> >
> > Thank you for the reply.
> >
> > 1. The pretrained diffusion models I mentioned in my review is the model that adopted to generate data since diffusion models, 1-rectified flow,  flow matching are basically different perspectives to understand the same thing. I am not ask to compare other diffusion-based generative models.
> >
> > 2. As you mentioned, the precision and recall will be reported by November 25. It is already November 25, but  it seems that these metrics have  not been reported so far.
> >
> > 3. Since no theoretical analysis can be given, it leaves your experimental results important.
> >
> > Overall, could you please provide your promised experimental results and more empirical evidence? Other reviewers also mentioned that the paper lacks evaluations and has no theoretical insights.

---

> ### Author Response · Authors · 2024-11-26
> **Response to Reviewer et3P(2)**
>
> Thank you once again for your thoughtful feedback and suggestions. We apologize for not being able to provide the additional materials promptly.
>
> $\textbf{[1]}$\
> Apologies for misunderstanding the intention of your first question. We agree with your suggestion that clarifying the position of 1-rectified flow within the context of flow matching-based generative models would indeed be helpful for readers.\
> Unfortunately, due to space constraints in the main manuscript, we could not include additional diffusion-based generative models in Table 1.\
> Instead, we have addressed this limitation by reporting the performance of as many diffusion-based generative models as possible in Appendix J, including your suggested models.\
> In particular, we have included the following representative models:
> * Diffusion-based generative models: EDM [1], LSGM [2], PFGM [3]
> * Distillation models: PD [4], DFNO [5], SID [6]
> * Consistency models: CD [7], CT [7], ICT [8], CTM [9], CTM+GAN [9]
>
> Additionally, we have clarified this point in Table 1 and  Section 4.1 (L357) as follows:
> >Further results clarify the upper bounds of reflow. We provide the CIFAR-10 unconditional generation qualities of pre-trained diffusion models in Appendix J.
>
> Thank you again for your valuable suggestions.
>
>
> $\textbf{[2,3]}$\
> Thank you again for your thoughtful comments. The following additional materials have been included:
> 1. $\textbf{Slerp Patterns and Lerp (Appendix B)}$\
> To provide further experimental insights into the use of Slerp and the noise patterns that increase and then decrease, we conducted additional experiments with the following settings:
> * Strictly increasing noise (SI) : Noise increases from 0.006 to 0.13.
> * Strictly decreasing noise (SD) :  Noise decreases from 0.13 to 0.006.
> * Using Lerp : Linear interpolation.
>
> The strictly increasing pattern allows us to analyze generation quality
> as noise progressively grows, while the strictly decreasing pattern helps us evaluate performance as
> noise is gradually removed.
>
> FID : $\textbf{6.63}$ (Ours) / 6.64 (SI) / 6.70 (SD) / 7.50 (Lerp)\
> IS : $\textbf{8.72}$ (Ours) / 8.48 (SI) / 8.45 (SD) / 8.46 (Lerp)
> - Our scheduling achieves higher FID and IS compared to other settings.
> - Using Slerp instead of Lerp generally results in higher FID and IS.
>
> 2. $\textbf{Recall and Precision (Appendix C)}$\
> Recall and precision for 2- and 3-rectified flow have been added, showing significant improvement in recall for 1-step sampling compared to the baseline.
>
> $\textbf{Multi step sampling(NFE=104)}$
> - 2-rectified flow\
> 	Precision : 0.696 (+0.005) / 0.691(ours)\
> recall :0.600 / 0.605 (+0.005) (ours)
> - 3-rectified flow\
> 	Precision : 0.698 (+0.007) / 0.691(ours)\
> recall : 0.592 / 0.599 (+0.007)(ours)
>
> $\textbf{1-step sampling}$
> - 2-rectified flow\
> 	Precision : 0.695 / 0.687 (+0.008) (ours)\
> recall :0.528 / 0.583 ($\textbf{+0.055}$) (ours)
> - 3-rectified flow\
> 	Precision : 0.682 (+ 0.009) / 0.691 (ours)\
> recall : 0.562 / 0.592 ($\textbf{+0.03}$) (ours)
>
> It indicate that while the precision of our method is nearly identical to the baseline, its higher recall demonstrates superior coverage of the real data distribution.
>
> 3. $\textbf{ Extreme Number of Reflow Steps (k = 4) (Appendix D)}$
> - Our method consistently outperforms the baseline across general reflow steps, not just for 2 and 3.
> - 4-Rectified flow, NFE 100\
> 		IS 8.95 / 9.01 (Ours) , FID 4.49 / 4.19 (Ours)
> - 4-Rectified flow, NFE 1\
> IS 8.59 / 8.80 (Ours), FID 6.58 / 5.66 (Ours)
>
> For more details, please refer to the revised PDF, Appendix B, C, and D.\
> We hope the additional materials submitted have helped experimentally support the effectiveness of our method. Any further questions or concerns would be gratefully addressed. Thank you.
>
>
> [1] Karras et el., Eucidating the design space of diffusion-based generative models, 2022.\
> [2] Vahdat et el., Score-based generative modeling in latent space, 2021.\
> [3] Xu et el., oisson flow generative models, 2022.\
> [4] Salimans & Ho, Progressive distillation for fast sampling of diffusion models, 2022.\
> [5] Zheng et el., Fast sampling of diffusion models via operator learning, 2023.\
> [6] Zhou et el., Score identity distillation: Exponentially fast distillation of pretrained diffusion models for one-step generation,2024.\
> [7] Song et el., Consistency models, 2023.\
> [8] Song & Dhariwal., Improved techniques for training consistency models, 2023.\
> [9] Kim et el., Consistency trajectory models: Learning probability flow ode trajectory of diffusion, 2023.

---

> ### Comment · Reviewer_et3P · 2024-12-02
>
> I have reviewed the authors’ rebuttal, and most of my concerns have been addressed.
>
> I appreciate the efforts the authors put into the rebuttal. However, considering the contribution level—such as the lack of theoretical analysis, the experimental scale, and the impact on future research—I am unable to raise my score to 8 and will **maintain it at its current level for weak acceptance**.

---

> > ### Author Response · Authors · 2024-12-02
> > **Response to Reviewer et3P(3)**
> >
> > Thank you for your thoughtful review and for acknowledging that most of your concerns have been addressed.
> >
> > Thank you for acknowledging the contributions of our empirical findings in demonstrating the drift phenomenon in the reflow process with real images. While we focused on empirical validation in this work, we see the development of a more theoretical approach as a promising direction for future research.
> >
> >  Additionally, we will provide results on ImageNet 64x64 and other complex datasets in the camera-ready version to strengthen our evaluation.
> >
> > We sincerely appreciate your constructive feedback, which has been invaluable in refining our work.

---

### Official Review · Reviewer_3L7y · 2024-11-02

**Soundness:** 3
**Presentation:** 3
**Contribution:** 3
**Rating:** 6
**Confidence:** 3

**Summary:**

The paper begins by identifying two key issues with current rectified flow models: (1) high computational costs due to the large synthetic datasets required for the reflow process, and (2) degradation in image quality with k-rectified flow, where generated images drift from the real data distribution, potentially leading to potential mode collapse. To address these challenges, the authors propose incorporating real images into the k-rectified flow process. This integration significantly reduces computational costs, as fewer samples are needed when using real images (by their experiments). The authors construct noise-image pairs by inverting real images and stabilize the process by blending the inverted and Gaussian noise through spherical interpolation.

Additionally, the paper introduces several enhancements, including balancing between traditional (synthetic-only) and real image-based rectified flows and implementing a Slerp noise schedule. They also propose a new metric, Initial Velocity Delta (IVD), to analyze model performance in one-step denoising. The proposed method demonstrates improvements over traditional k-rectified flow models, as shown through evaluations on CIFAR-10.

**Strengths:**

The authors begin with a thoughtful analysis of the limitations in existing rectified flow models, which effectively motivates their proposed approach. Their decision to incorporate real images into the reflow process is both intuitive and straightforward to implement. Additionally, they smartly apply spherical interpolation to blend Gaussian and inverted noise, promoting stable training, and introduce balancing techniques to integrate synthetic-only and real-data flows for greater consistency. The introduction of the IVD metric further enhances the paper’s novelty by offering an innovative tool for in-depth analysis of one-step denoising, providing valuable insights into model behavior and effectiveness, especially in assessing the initial accuracy of generative paths.

**Weaknesses:**

Although the paper presents substantial improvements to the k-rectified flow process, a more in-depth theoretical foundation could enhance the rigor and clarity of the contributions. Some key points to consider:
1. The use of real images in reflow may seem to contradict the original purpose of the rectified flow method. Since reflow is based on the fact that it used the new synthesized data which was rewired, making it easier to straighten the path. Real data does not carry that property, so integrating real images might theoretically undermine the reflow process. While experimental results indicate positive outcomes, a theoretical examination of why real images enhance rather than disrupt the reflow process would strengthen the study and clarify this counterintuitive result.
2. The choice of spherical interpolation (Slerp) lacks adequate justification. It would be helpful to understand why Slerp was selected over simpler alternatives like linear interpolation. Is there a theoretical or empirical reason why Slerp provides a significant advantage for stability or image quality in this context? A comparison with other interpolation methods could demonstrate whether Slerp offers unique benefits.
3. Similarly, the noise scheduler’s reliance on spherical interpolation, with a gradual increase and then decrease in Gaussian noise, is not fully explained. An explanation for this scheduling pattern, ideally supported by theoretical or experimental evidence, would clarify its role in enhancing training or image quality and differentiate it from simpler approaches.
4. Since the method stated that it should helps the model avoiding mode collapse, it would be nice to see even more extreme k (papers examined till k=3) to see the effectiveness of the method.

**Questions:**

Authors should address the questions raised in Weakness section.

---

> ### Author Response · Authors · 2024-11-21
> **Response to Reviewer 3L7y**
>
> We appreciate your thoughtful and detailed feedback on our work. We hope these explanations resolve the concerns, and we would greatly appreciate any further questions or suggestions.
>
>
> $\textbf{[W1] property of $v^{-1}(X_1)$}$\
> Thank you for raising your concern.\
> Using real images does not contradict the original purpose because the noise $v^{-1}(X_1)$ generated using $ v^{-1} $ satisfies the properties of the rectified flow as well.
>
> Moreover, the original paper also mentions that $ v^{-1} $ (or $-v$) can be used optionally, supporting the validity of this approach. To clarify this point further, we have added explanations about $v^{-1}$ and $v^{-1}(X_1)$ in lines 191–193. Thank you for raising this insightful point.
>
> $\textbf{[W2] The reason for Slerp}$\
> Interpolation in noise space using Slerp is a common practice in generative models ([1],  [2]).\
> We agree that clarifying the use of Slerp and comparing it with other methods adds value. We have added an explanation with citations in Lines 233–235 and will include Lerp results in Appendix C by November 25. \
> Thank you for your feedback.
>
> $\textbf{[W3] About Slerp scheduling pattern}$
>
> As noted in the Limitations section, the noise scheduling used with Slerp is heuristic and based on empirical experimentation, not a theoretical foundation. While effective, it may not be optimal.
>
>  We agree that analyzing various noise scheduling patterns (e.g., monotonically decreasing, increasing) would provide valuable insights and will include additional results in the appendix by November 25. Thank you for your suggestion.
>
> $\textbf{[W4] The impact of ours when $K > 3$}$\
> Thank you for the suggestion. While rectified flow methods typically use $k=2$ or $k=3$, exploring $ k=4 $ or higher could evaluate our method’s robustness against model collapse.
>
> Running $ k=5 $ experiments is challenging within the discussion period because $ k=4 $ training must be completed first to generate new pairs. We will include $ k=4 $ results in the appendix B by November 25.
>
>
> Thank you again for the constructive comments. We will report the results in the appendix once the additional experiments are completed. Your feedback has enriched our work.
>
>
> [1] Wang & Golland., 2023, Interpolating between images with diffusion models.\
> [2] Jang et al., 2024, Spherical linear interpolation and text-anchoring for zero-shot composed image retrieval.

---

> > ### Comment · Reviewer_3L7y · 2024-11-26
> >
> > Thank you, authors, for providing clarification. Your thorough response has addressed all of my concerns, and I am now confident in my decision to maintain the score as it stands. I appreciate your time and effort in resolving these matters.

---

> > > ### Author Response · Authors · 2024-11-29
> > > **Response to Reviewer 3L7y (2)**
> > >
> > > Thank you for your thoughtful and positive feedback.
> > >
> > > We appreciate your time and effort in reviewing our work and are glad that our clarifications have addressed your concerns.

---

### Official Review · Reviewer_Hw3H · 2024-11-03

**Soundness:** 2
**Presentation:** 1
**Contribution:** 2
**Rating:** 3
**Confidence:** 3

**Summary:**

This paper claimed that rewiring the flow starting from the source distribution (e.g. the Gaussian distribution) to the real data distribution in rectified flow generates the drifted distribution. To remedy this, the authors proposed to start from the real data distribution for rewiring. After obtaining the inversion of each real data, the authors use the spherical linear interpolation (slerp) between the reverse noise and a randomly sampled noise. The training objective involves data from both inversion directions.

**Strengths:**

The authors did analysis investigating the rewiring directions in the original rectified flow. There might be a distribution drifting phenomenon that we should pay attention to.

**Weaknesses:**

The authors make some claims without strong evidence:
- The authors mentioned in line 124: "Interestingly, the k-rectified flow underperforms the (k − 1)-rectified flow in terms of image quality. This is obvious because the fake samples have lower quality than the real samples". This sounds too strong. Won't there be other reasons?
- The authors mentioned that "Since the reflow process uses only fake pairs, discrepancies occur between the reconstruction errors of real and generated images." However, it is still not obvious why reflow process uses only fake pairs is a reason for "discrepancies occur between the reconstruction errors of real and generated images"
- Why "$L^{recon}_2$ is lower at the fake samples than the real samples" indicates "the 2-rectified flow drifts away from the real samples" (line 151)?
- Why "Lp-recon is lower near the fake samples than the real samples" indicates "the 2-rectified flow suffer from crossing between real samples"  (line 152)?
- The authors mentioned that the original Rectified Flow algorithm (Liu et al. 2022) uses the fake pairs, but I have to mention that the original Rectified Flow algorithm provided an option to use the real data pairs $(v^{-1}(X_1), X_1)$.

Experimental results contradict with the claims:
- From the experiments in Fig. 3(b), the perturbed reconstruction error for real samples are higher than the reconstruction error for real samples. This is not faithful. This contradicts with the perturbation proposed in the method in Sec. 3.3
- This paper discussed that in the original rectified flow, more number of rectified flow lowers the image quality. Therefore, the authors proposed to use real pairs. However the proposed method also suffers from more number of rectified flows being worse (Figure 7).

Many parts in this paper is not clear and confusing:
- The generated pair or fake pair in this paper seems confusing. Please have formal definitions of the generated pair, fake pair, real pair.
- What are fake samples? Are those data generated from the source distribution (e.g., Gaussian distribution) via a trained $v$?
- In line 141 the authors wrote: "Lower reconstruction error on the real samples $_2^{recon}(X_1)$ indicates that a rectified flow is more faithful to the real samples as shown in Figure 3b." However, in Fig. 3(b), real samples have higher reconstruction error. I got confused about what the authors want to express.
- Definition of exp() in Eq. (10) and (12). Is it an exponential function? I understand $t$ is generally between 0 and 1.
- In line 295, this method needs to define a slerp schedule with respect to time t. How is the time t in line 295 related to training steps in Fig. 5? How to set the proportional coefficient for the slerp noise schedule?
- What does the "single real pair" in "reflow using just a single real pair" in Sec. 4.6 Ablation study refer to?
- Table 1's results are not explained.

Experimental results are not strong:
- From the ablation study, the method without slerp noise performance is similar to the proposed method, in terms of Inception Score, 8.79 vs 8.57. The authors should do experiments on more datasets to evaluate the effectiveness of the proposed real pair and slerp noise.
- No quantitative comparison between the proposed method and the baseline method on the LSUN bedroom dataset. In terms of the qualitative results, when the number of steps is 2 (Fig. A27) or higher (Fig. A28), the results of the proposed method and the baseline method appear close.

**Questions:**

See Weakness

---

> ### Author Response · Authors · 2024-11-22
> **Response to Reviewer Hw3H**
>
> Thank you for your detailed and thoughtful feedback. We have thoroughly considered your suggestions and integrated them into our work. As there is a lot to address, I will provide my responses in two parts.
>
> $\textbf{[W1,2]}$\
> Thank you for raising your concern.\
> Fortunately, it is not too strong as follows. $v_1(X_0)$ drifts away from $X_1$. $v_2(X_0)$ drifts away from $v_1(X_0)$ where $v_2$ is 2-rectified flow. Among the possible choices of $v_2(X_0)$, the possibility of being closer to $X_1$ than $v_1(X_0)$ is very small.\
> Likewise, discrepancies between the reconstruction errors of real and fake images is obvious because the fake samples are not equal to the real samples. The optimal solution of $v_2$ with the fake samples as supervision is $v_2(X_0)=v_1(X_0)$, not $v_2(X_0) = X_1$.
>
> $\textbf{[W3]}$\
> Thank you for raising your curiosity.
> * Reconstruction error measures the drift from the original samples.
> * The 2-rectified flow makes higher reconstruction error on the real samples than the fake samples.
> * Hence, it indicates the 2-rectified flow drifts away from the real samples.
>
> We suppose that the order of the comparative brought confusion on this simple syllogism. We have fixed the sentence to be:
> >$L_2^\text{recon}$ is higher at the real samples than the fake samples.(L144)
>
>
> $\textbf{[W4]}$\
> Thank you for raising your curiosity.\
> Figures 2 and 3(b) suggest that the noise space for fake samples is well-structured, but the manifold for real samples remains unstable, making them more sensitive to perturbations. As a result, the model captures fake sample structure effectively but struggles with real samples.
>
> Thank you for raising your concern
>
>
> $\textbf{[W5]}$\
> As correctly noted, the original rectified flow provides the option to use reverse trajectories for pairing, and our method builds on this.
> However, while the original rectified flow just mentions that reverse trajectories can be inferred using $-v$, Our approach goes further by incorporating both real and fake pairs into the reflow process and applying Slerp to $v^{-1}(X_1) $.
>
> The advantages of Slerp are as follows:
>
> * Slerp helps maintain the trajectory and neighborhood between reverse noise corresponding to real images more stably. This reduces $L_2^{recon}$ and $L_2^{p-recon}$ errors for real images.
>
> * Better generation quality(FID, IS) than no-Slerp settings.
>
> To clarify, we have added a note in Line 192 of the revised paper acknowledging the original paper's introduction of this option.
>
>
> $\textbf{[W6]}$\
> Thank you for raising your concern.
> * The perturbed reconstruction error for real samples measures the reconstruction error of perturbed versions of real samples.
> * Our training method perturbs the reversed noise from real images, not the images.
>
> Hence, they do not contradict each other.
>
>
> $\textbf{[W7]}$\
> Thank you for the insightful observation.\
> This issue stems from a fundamental limitation shared by rectified flow methods that rely on fake images during the reflow process.
> * We have added this limitation in Lines 526–530 of the revised paper.
>
> Nevertheless, it is important to note that our method improves the original reflow process, which is the core contribution of this work.\
> Thank you for your valuable feedback.
>
> $\textbf{[W8,9]}$\
> Thank you for raising your concern.\
> We use "fake pair" and "generated pair" interchangeably, while "real pair" refers specifically to real images $X_1$​ and $v^{-1}(X_1)$.  However, we acknowledge that using terms like "generated image" without clarification after defining "fake pair" might cause confusion for readers.\
> We have made the following revisions:
> * Replaced "generated image" with "fake image" (Line 416, Figure 3(b) + caption, Line 415).
> * Added further clarification about "fake," "real," and "generated" in Lines 197–199.
>
> Thank you for the insightful suggestion.
>
>
> $\textbf{[W10]}$\
> We apologize for the confusion.\
> The referenced part should have pointed to Figure 9(b) rather than Figure 3(b). This was an error.
>
> As noted in Lines 151–154 of the revised version, the current rectified flow shows lower reconstruction errors for fake pairs than real pairs, indicating a bias toward fake pairs in the original reflow methods. Our claim is :
> > "Our rectified flow achieves lower reconstruction error for real samples than the original rectified flow."
>
> We decided to remove L141 to avoid redundancy and confusion.\
> Thank you for pointing this out.
>
> $\textbf{[W11]}$\
> Thank you for raising your concern.\
> Range of $p_{t}(u)$ is also $[0,1]$.
> For further details, a visualization of $p_{t}(u)$ is provided in Appendix A.\
> We hope this clarifies your question.
>
>  (to be continued)

---

> ### Author Response · Authors · 2024-11-22
> **Response to Reviewer Hw3H(2)**
>
> $\textbf{[W12]}$\
> We apologize for the confusion.\
> The scheduling of Slerp depends on $\zeta \in [0, 1]$ --not depend on $t$, and when the maximum $\zeta_{max}$ is fixed (e.g., 0.006), it decreases towards 0 as the training step progresse( Fig 5 ).
>
> I apologize for the confusion caused by representing the Slerp schedule in terms of $t$.
> The corrected is provided in Revision PDF L301 :
>
> * $\text{slerp schedule}(\zeta) \propto 1 - \frac{{\zeta}^2}{1 + {\zeta}^2}, \zeta \in [0, 1].$
>
> The maximum noise $\zeta_{max}$ is simply scaled linearly, increasing from 0.006 to 0.13 and then decreasing. For example, $\zeta_{max}^{1} = 0.006 \times 1$ to $\zeta_{max}^{4} = 0.13 \times 1$  (Fig 6).\
> We have incorporated the following revisions into the updated PDF for clarification:
> * Lines 298–299: Added explanation for Figure 5.
> * Lines 305–308: Added explanation for Figure 6.
> * Figures 5 and 6: Adjusted maximum noise to align with the values in the paper.\
> We apologize for the oversight and appreciate your feedback
>
> $\textbf{[W13]}$\
> Thank you for raising your curiosity.\
> This setting fixes the real pair just once and does not continuously update real pairs through repairing.
>
> The ablation study shows that continuously updating real pairs achieves better generation quality than a single pair because it helps preserve the velocity field of the rectified flow for real images.
>
> To provide a more detailed explanation of the ablation settings, we have moved this explanation from Appendix A to the main paper in Lines 504–507.
>
> Changed its name to avoid ambiguity:
> * Single Real Pair => Fixed Real Pair
>
> $\textbf{[W14]}$\
> Thank you for raising your curiosity.\
> Table 1 implies that :
>
> * Better quality in all steps.
> * Outperforms even with lower NFE => Indicates reduced drift away from real images in the reflow process.
> * Superior with the same distillation method.
>
>
> Added further explanations in the updated PDF (Lines 350–352), stating:
>
> >"Furthermore, our method achieves better generation quality in RK sampling, despite having lower NFE than the baseline. This indicates reduced drift away from real images in the reflow process.”
>
> We appreciate your valuable feedback.
>
> $\textbf{[W15]}$\
> Thank you for your question.\
> Although the IS values show little difference:
>
> * FID significantly improves with Slerp over no-Slerp.
> * Lower $L_2^{\text{recon}}(X_1)$ and $L_2^{\text{p-recon}}(X_1)$ than no-Slerp settings.
>
> This shows the benefits of Slerp.\
> Unfortunately, generating 4M fake pairs and retraining the rectified flow on another dataset is infeasible within the discussion period. We apologize for the limitation.
>
> $\textbf{[W16]}$\
> Thank you for your suggestion.\
> The 2-rectified flow trained on the LSUN dataset used only 120k fake pairs for training.
> This is a significantly smaller number of fake images in the original proposal.
> Fair FID comparison with the baseline requires at least 4M LSUN fake pairs, which is not feasible within the discussion period.\
> Instead, we show that our method is significantly better than the baseline in various quantitative metrics :
> * IVD(Initial Velocity Delta)
> * $L_2^{recon}$ and $L_2^{p-recon}$ of real images.
> *  $L_2^{recon}$ and $L_2^{p-recon}$ of fake images.\
> Table A1 in Appendix C.
>
>
>
> Thank you once again for the great interest in our work and the valuable suggestions. If there are any further thoughts or suggestions, they would be greatly appreciated.

---

### Official Review · Reviewer_7ckE · 2024-11-03

**Soundness:** 3
**Presentation:** 2
**Contribution:** 3
**Rating:** 5
**Confidence:** 3

**Summary:**

This paper presents Balanced conic rectified flow, an improved version of "reflow" in rectified flow. The main idea is to use a real sample and its reverse Gaussian random noise pair based on inverse flow rather than using noise and corresponding generated samples using PF ODE. To mitigate the error from the inverse flow, the paper presents a "conic" reflow that considers a neighbor of the inverse point with a small noise. The paper argues the proposed method shows better $k$-rectified flow for $k>1$, especially for 1-step generation.

**Strengths:**

- To the best of my knowledge, using real data and its reverse Gaussian random noise is an interesting and under-explored direction.
- The performance improvement on CIFAR-10 is promising.
- The observations (Figures 2 and 3) that motivate the method are interesting.

**Weaknesses:**

I am slightly negative to the paper due to the clarity/readability problem. Specifically:

- The writing of the paper can be improved. For instance, since the primary focus of the paper is the transport from Gaussian distribution to the target data distribution, explicitly mentioning it (like $X_1$ or $X_0$ becomes the real data and the Gaussian noise, respectively), can help the readers to understand the contents better, especially for the people who are not very familiar with the concept of Rectified Flow. Moreover, some captions are overlapped with other figures, such as in Figures 2 and 3. Other figures are really problematic; for instance, in Figure 9 and 10, legends and labels are almost impossible to read.
-  The paper compares the performance only with the original rectified flow and does not provide a comparison with relevant work [1]. Although it may be understandable because the primary focus of this paper is in revisiting the "reflow" process, I think the quantitative comparison should exist because this work also deals with the same problem. Otherwise, could authors try conic reflow upon this paper?
- As mentioned in the paper, the paper lacks theoretical analysis when choosing slerp or conic hyperparameters. In this respect I expect more empirical results (like imagenet or various higher-resolution datasets) should be provided, but the in the current form the paper only provides retuls on CIFAR-10 and (qualitative) evaluation on LSUN. Does a similar trend hold for AFHQ or other datasets? I think the paper should include the results at least on all datasets used in the original rectified flow datasets, given that the original paper provides all of the checkpoints for those datasets and this paper does not have theoretical insights (meaning that the validation should be done in empirical manner).
- The values in Table 2 are difficult to understand. What do they mean?
- Suggestion: do we need to write the reverse ODE as $v^{-1}$? Isn't it just $-v$?
- Minor: curvature -> Curvature in L410.

[1] Lee et al., 2024  Improving the Training of Rectified Flow

**Questions:**

- In Table 3, why does NFE between the original rectified flow and the proposed method differ (like 110 vs. 104)?
- Did authors try using only real pairs for reflow (without fake pairs)?

---

> ### Author Response · Authors · 2024-11-21
> **Response to Reviewer 7ckE**
>
> Thank you for your constructive feedback. We have carefully reviewed your comments and have incorporated them into our work.
>
> $\textbf{[W1] Comments of $X_0$ and $X_1$}$ \
> Thank you for your thoughtful suggestion. We have added L093 (marked in red) to clarify $X_0$ and $X_1$ as below.
> > In image generation tasks, $X_0\sim\pi_0$ and $X_1\sim\pi_1$ are latent noises and real images, sampled from Gaussian distribution and data distribution, respectively.
>
> $\textbf{[W2] Issues with figures}$\
> We apologize for the inconvenience and thank you for finding them out. We updated them as follows.
>
> $\bullet$ The overlap between Figures 2 and 3 is resolved.\
> $\bullet$ The axes and legends are written with larger fonts in Figures 7, 8 ,9, 10(a), 10(b), 11 and 12.
>
> $\textbf{[W3] Quantitative comparison with [1] \}$\
> Thank you for the insightful comment.\
> We primarily compared with the original rectified flow rather than other diffusion-based generative models because our focus is issues due to fake pairs in the reflow process. \
> Our focus was on addressing issues with using only fake pairs in the reflow process, so we primarily compared with the original rectified flow rather than other diffusion-based generative models. \
> Similarly, we excluded [1] because its key points are orthogonal to ours:\
> $\bullet$ using EDM weight for pairing fake pairs\
> $\bullet$ using Huber loss for reflow\
> $\bullet$ Developing the sampling distribution for linear interpolation between $X_0$ and $X_1$.
>
> Although we agree that comparison with [1] would enrich our paper, it is difficult to compare as follows. We cannot directly compare the measures in [1] because its training configurations differ \
> (batch size: ours : 256 /  Lee : 512; training iterations: ours : 600K / Lee : 800K) from ours.
>
> We cannot train ours on their configuration or theirs in our configuration because it takes more than two weeks due to:
>
>
> $\bullet$ [1] uses a diffusion-based model with EDM-initialized weights instead of a 1-rectified flow model for pairing.\
> $\bullet$ Generating millions of pairs with [1]'s configuration.\
> $\bullet$ Training for several proposed losses applying our methods.
>
> We have added the keypoints of [1] and potential combination in Line 535~537. Which is infeasible within the discussion period. \
> Thank you for the suggestion.
>
>
> $\textbf{[W4] Experiments on other datasets}$\
> Unfortunately, the official repo only provides 1-rectified flow checkpoints, and training a comparable 2-rectified flow requires 4M fake pairs, which is infeasible during the rebuttal period.
>
> Instead, we trained on LSUN with the original 2-rectified flow to provide both qualitative and quantitative results including recon and p-recon errors, IVD, and curvature, as shown in Appendix C. We plan to train additional checkpoints from scratch and share them in the camera ready.
>
> $\textbf{[W5] Table2}$\
> Fine-tuned version has lower curvature and IVD than the original, indicating that our method is more straight than the original. And also fine-tuned version has lower recon and p-recon differences between real and fake images, indicating that our method reduces the bias toward fake samples.
>
> To visually show the performance gaps, we have replaced Table 2 with Figure 11 and clarified this in the revised PDF (Lines 449–451). Thank you.
>
> $\textbf{[W6]} v^{-1}$ and $-v$\
> Thank you for your remark. We used $v^{-1} $ to concisely represent the reverse noise for a real pair, allowing for a more general and compact notation. Since $ v^{-1} = -v $ in our case, we have added a footnote (Line159) in the revised PDF. We believe this addition improves the paper’s clarity.
>
> $\textbf{[W7]}$"curvature" in L410.\
> Thank you for pointing this out. We have corrected it.
>
> $\textbf{[Q1] NFE (110 vs 104)}$\
> The NFE difference comes from RK sampling adaptively adjusting based on a threshold. We averaged NFE over 50k samples but did not explicitly mention this, which may have caused confusion.\
> $\bullet$ Add details about the Runge-Kutta method in the revised paper (Lines 323, 344) and citation for tolerances and parameter settings.
>
> Thank you for your feedback.
>
> $\textbf{[Q2] Using only real pairs}$\
> Yes, we have tested.  Similar results to those observed in Appendix B show that this improves 1-step generation but not many-step generation, likely because real images alone cannot fully rewire the velocity field of 1-rectified flow.
>
> This raises important questions about the optimal real-to-fake pair ratio and the feasibility of straightening the velocity field with only real pairs, which are promising directions for future research. Thank you for your insightful question.
>
> \
> Thank you for providing a detailed response and good suggestions. Your feedback has been helpful in clarifying the content of the paper and addressing problematic figures. We hope that the issues you pointed out have been addressed and would greatly appreciate any further thoughts or suggestions you may have.

---

> > ### Comment · Reviewer_7ckE · 2024-11-25
> > **Response**
> >
> > Thanks for your detailed response. However, most of my concerns are not addressed. First, many figures are still problematic; for instance, the fonts in Figure A2 are too small to read. More importantly, since the paper does not provide any theoretical insight, I think the paper should try more extensive experimental evaluation --- like higher resolution (in addition to LSUN) or more complex (e.g., ImageNet 64x64). To sum up, I think the idea is good, but its soundness is yet weak due to a lack of evaluations and no theoretical insights. For me, either one of them should be provided to raise the score. Thus, I decide to retain my score.

---

> > > ### Author Response · Authors · 2024-11-29
> > > **Response to Reviewer 7ckE (2)**
> > >
> > > Thank you for your thoughtful and detailed feedback.
> > >
> > > We appreciate your comments on the problematic Figure A2 and are grateful for pointing them out. We have corrected the issues with the figures, including improving font sizes and ensuring their readability.
> > >
> > > Additionally, we value your detailed suggestion for a more extensive experiment (e.g., ImageNet 64x64) and fully acknowledge its importance. We will make sure to include results on ImageNet 64x64 and other complex datasets in the camera-ready version to strengthen our evaluation.
> > >
> > > Once again, thank you for your detailed and constructive feedback. It has been invaluable in improving our work.

---

### Meta-Review · Area_Chair_pWZq · 2024-12-21

**Metareview:**

This paper proposes to improve rectified flow models by incorporating real samples into the reflow process with the idea of leveraging the inverse flow. The strengths of this paper: The reviewers value the novelty and the contribution of this paper. The weaknesses of this paper: the paper's writing needs improvement, and the reviewers had concerns with the experimental comparisons. After the discussion period, two of the reviewers remained unconvinced and suggested conducting more evaluation and rewriting.

**Additional Comments On Reviewer Discussion:**

During the discussion period, reviewers did respond to the authors; however, the response did not fully address the concerns of reviewers 7ckE and Hw3H. Specifically, the presentation of the paper should be improved and Reviewer 7ckE believes that a more comprehensive evaluation of the approach is necessary.

---

### Decision · Program_Chairs · 2025-01-22

Reject